# Temporally Disentangled Representation Learning

**Weiran Yao**
CMU
weiran@cmu.edu

**Guangyi Chen**
CMU & MBZUAI
guangyichen1994@gmail.com

**Kun Zhang**
CMU & MBZUAI
kunz1@cmu.edu

## Abstract

Recently in the field of unsupervised representation learning, strong identifiability results for disentanglement of causally-related latent variables have been established by exploiting certain side information, such as class labels, in addition to independence. However, most existing work is constrained by functional form assumptions such as independent sources or further with linear transitions, and distribution assumptions such as stationary, exponential family distribution. It is unknown whether the underlying latent variables and their causal relations are identifiable if they have arbitrary, nonparametric causal influences in between. In this work, we establish the identifiability theories of nonparametric latent causal processes from their nonlinear mixtures under fixed temporal causal influences and analyze how distribution changes can further benefit the disentanglement. We propose **TDRL**, a principled framework to recover time-delayed latent causal variables and identify their relations from measured sequential data under stationary environments and under different distribution shifts. Specifically, the framework can factorize unknown distribution shifts into transition distribution changes under fixed and time-varying latent causal relations, and under observation changes in observation. Through experiments, we show that time-delayed latent causal influences are reliably identified and that our approach considerably outperforms existing baselines that do not correctly exploit this modular representation of changes. Our code is available at: https://github.com/weirayao/tdrl.

## 1 Introduction

Causal reasoning for time-series data is a fundamental task in numerous fields [1, 2, 3]. Most existing work focuses on estimating the temporal causal relations among observed variables. However, in many real-world scenarios, the observed signals (e.g., image pixels in videos) do not have direct causal edges, but are generated by latent temporal processes or confounders that are causally related. Inspired by these scenarios, this work aims to uncover causally-related latent processes and their relations from observed temporal variables. Estimating latent causal structure from observations, which we assume are unknown (but invertible) nonlinear mixtures of the latent processes, is very challenging. It has been found in [4, 5] that without exploiting an appropriate class of assumptions in estimation, the latent variables are not identifiable in the most general case. As a result, one cannot make causal claims on the recovered relations in the latent space.

Recently, in the field of unsupervised representation learning, strong identifiability results of the latent variables have been established [6, 7, 8, 9, 10] by using certain side information in nonlinear Independent Component Analysis (ICA), such as class labels, in addition to independence. For time-series data, history information is widely used as the side information for the identifiability of latent processes. To establish identifiability, the existing approaches enforce different sets of functional and distributional form assumptions as constraints in estimation; for example, **(1)** PCL [7], GCL [8], HM-NLICA [11] and SlowVAE [12] assume mutually-independent sources in the data generating process. However, this assumption may severely distort the identifiability if the

36th Conference on Neural Information Processing Systems (NeurIPS 2022).

Table 1: Attributes of nonlinear ICA theories for time-series. A check denotes that a method has an attribute or can be applied to a setting, whereas a cross denotes the opposite. † indicates our approach.

| Theory | Time-varying Relation | Causally-related Process | Partitioned Subspace | Nonparametric Transition | Applicable to Stationary Environment |
|---|---|---|---|---|---|
| PCL | ✗ | ✗ | ✗ | ✓ | ✓ |
| GCL | ✓ | ✗ | ✗ | ✓ | ✓ |
| HM-NLICA | ✗ | ✗ | ✗ | ✗ | ✗ |
| SlowVAE | ✗ | ✗ | ✗ | ✗ | ✓ |
| SNICA | ✓ | ✓ | ✗ | ✗ | ✗ |
| i-VAE | ✓ | ✗ | ✗ | ✗ | ✗ |
| LEAP | ✗ | ✓ | ✗ | ✓ | ✗ |
| **TDRL** † | ✓ | ✓ | ✓ | ✓ | ✓ |

latent variables have time-delayed causal relations in between (i.e., causally-related process); **(2)** SlowVAE [12] and SNICA [13] assume linear relations, which may distort the identifiability results if the underlying transitions are nonlinear, and **(3)** SlowVAE [12] assumes that the process noise is drawn from Laplacian distribution; i-VAE [9] assumes that the conditional transition distribution is part of the exponential family. However, in real-world scenarios, one cannot choose a proper set of functional and distributional form assumptions without knowing in advance the parametric forms of the latent temporal processes. Our first step is hence to understand under what conditions the latent causally processes are identifiable if they have nonparametric transitions in between. With the proposed condition, our approach allows recovery of latent temporal causally-related processes in stationary environments without knowing their parametric forms in advance.

On the other hand, nonstationarity has greatly improved the identifiability results for learning the latent causal structure [14, 15, 16]. For instance, LEAP [14] established the identifiability of latent temporal processes, but in limited nonstationary cases, under the condition that the distribution of the noise terms of the latent processes varies across all segments. Our second step is to analyze how distribution shifts benefit our stationary condition and to extend our condition to a general nonstationary case. Accordingly, our approach enables the recovery of latent temporal causal processes in a general nonstationary environment with time-varying relations such as changes in the influencing strength or switching some edges off [17] over time or domains.

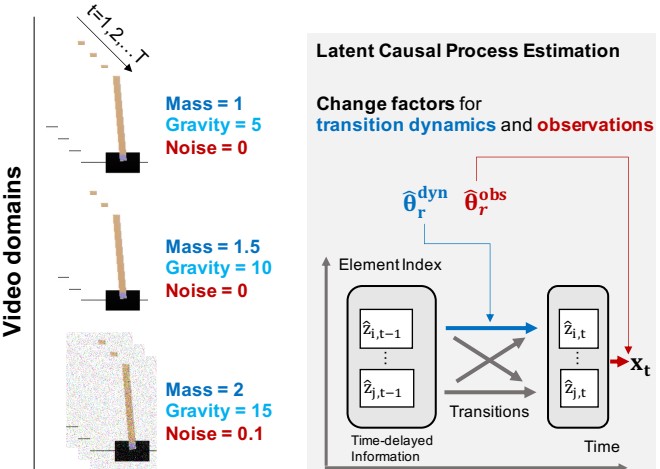

Figure 1: **TDRL**: Temporally Disentangled Representation Learning. We exploit fixed causal dynamics and distribution changes from changing causal influences and global observation changes to identify the underlying causal processes. $\hat{z}_{i,t}$ is the estimated latent process. $\hat{\theta}_r^{\mathrm{dyn}}$ is the change factor for **transition dynamics**, i.e., representing mass and gravity in this example. $\hat{\theta}_r^{\mathrm{obs}}$ is the change factor for **observation**, i.e., noise scale.

Given the identifiability results, we propose a learning framework, called **TDRL**, to recover nonparametric time-delayed latent causal variables and identify their relations from measured temporal data under stationary environments and under nonstationary environments in which it is unknown in advance how the joint distribution changes across domains (we define it as "unknown distribution shifts"). For instance, Fig. 1 shows an example of multiple video domains of a physical system under different mass, gravity, and environment rendering settings[1]. With **TDRL**, the differences across segments are characterized by the learned change factors $\hat{\theta}_r^{\mathrm{dyn}}$ of domain $r$ (note that domain index is given to the model) that encode changes in transition dynamics, and changes in observation or styles modeled by $\hat{\theta}_r^{\mathrm{obs}}$ (we use "causal dynamics" and "latent causal relations/influences" interchangeably). We then present a generalized time-series data generative model that takes these change factors as arguments for modeling the distribution changes. Specifically,

---

[1]The variables and functions with "hat" are estimated by the model; the ones without "hat" are ground truth.

we factorize unknown distribution shifts into transition distribution changes in stationary processes, time-varying latent causal relations, and global changes in observation by constructing partitioned latent subspaces, and propose provable conditions under which nonparametric latent causal processes can be identified from their nonlinear invertible mixtures. We demonstrate through a number of real-world datasets, including video and motion capture data, that time-delayed latent causal influences are reliably identified from observed variables under stationary environments and unknown distribution shifts. Through experiments, we show that our approach considerably outperforms existing baselines that do not correctly leverage this modular representation of changes.

## 2 Related Work

**Causal Discovery from Time Series**    Inferring the causal structure from time-series data is critical to many fields including machine learning [1], econometrics [2], and neuroscience [3]. Most existing work focuses on estimating the temporal causal relations between observed variables. For this task, constraint-based methods [18] apply the conditional independence tests to recover the causal structures, while score-based methods [19, 20] define score functions to guide a search process. Furthermore, [21, 22] propose to fuse both conditional independence tests and score-based methods. The Granger causality [23] and its nonlinear variations [24, 25] are also widely used.

**Nonlinear ICA for Time Series**    Temporal structure and nonstationarities were recently used to achieve identifiability in nonlinear ICA. Time-contrastive learning (TCL [6]) used the independent sources assumption and leveraged sufficient variability in variance terms of different data segments. Permutation-based contrastive (PCL [7]) proposed a learning framework which discriminates between true independent sources and permuted ones, and identifiable under the uniformly dependent assumption. HM-NLICA [11] combined nonlinear ICA with a Hidden Markov Model (HMM) to automatically model nonstationarity without manual data segmentation. i-VAE [9] introduced VAEs to approximate the true joint distribution over observed and auxiliary nonstationary regimes. Their work assumes that the conditional distribution is within exponential families to achieve the identifiability of the latent space. The most recent literature on nonlinear ICA for time-series includes LEAP [14] and (i-)CITRIS [26, 27]. LEAP proposed a nonparametric condition leveraging the nonstationary noise terms. However, all latent processes are changed across contexts and the distribution changes need to be modeled by nonstationary noise and it does not exploit the stationary nonparametric components for identifiability. Alternatively, CITRIS proposed to use intervention target information for identification of scalar and multidimensional latent causal factors. This approach does not suffer from functional or distributional form constraints, but needs access to active intervention.

## 3 Problem Formulation

### 3.1 Time Series Generative Model

**Stationary Model**    As a **fundamental** case, we first present a regular, stationary time-series generative process where the observations $\mathbf{x}_t$ comes from a nonlinear (but invertible) mixing function $\mathbf{g}$ that maps the time-delayed causally-related latent variables $\mathbf{z}_t$ to $\mathbf{x}_t$. The latent variables or processes $\mathbf{z}_t$ have stationary, nonparametric time-delayed causal relations. Let $\tau$ be the time lag:

$$\underbrace{\mathbf{x}_t = \mathbf{g}(\mathbf{z}_t)}_{\text{Nonlinear mixing}}, \quad \underbrace{z_{it} = f_i\left(\{z_{j,t-\tau}|z_{j,t-\tau} \in \mathbf{Pa}(z_{it})\}, \epsilon_{it}\right)}_{\text{Stationary nonparametric transition}} \; with \; \underbrace{\epsilon_{it} \sim p_{\epsilon_i}}_{\text{Stationary noise}} \; .$$

Note that with nonparametric causal transitions, the noise term $\epsilon_{it} \sim p_{\epsilon_i}$ (where $p_{\epsilon_i}$ denotes the distribution of $\epsilon_{it}$) and the time-delayed parents $\mathbf{Pa}(z_{it})$ of $z_{it}$ (i.e., the set of latent factors that directly cause $z_{it}$) are interacted and transformed in an arbitrarily nonlinear way to generate $z_{it}$. Under stationarity assumptions, the mixing function $\mathbf{g}$, the transition functions $f_i$ and the noise distributions $p_{\epsilon_i}$ are invariant. Finally, we assume that the noise terms are mutually-independent (i.e., spatially and temporally independent), which implies that instantaneous causal influence between latent causal processes is not allowed by the formulation. The stationary time-series model in the fundamental case is used to establish the identifiability results under fixed causal dynamics in Section 4.1.

**Nonstationary Model**    We further consider two violations of the stationarity assumptions in the fundamental case, which lead to two nonstationary time series models. Let $\mathbf{u}$ denote the domain

or regime index. Suppose there exist $m$ regimes of data, i.e., $u_r$ with $r = 1, 2, ..., m$, with unknown distribution shifts. In practice, the changing parameters of the joint distribution across domains often lie in a low-dimensional manifold [28]. Moreover, if the distribution is causally factorized, the distributions are often changed in a minimal and sparse way [29]. Based on these assumptions, we introduce the low-dimensional minimal change factor $(\boldsymbol{\theta}_r^{\mathrm{dyn}}, \boldsymbol{\theta}_r^{\mathrm{obs}})$, which was proposed in [30], to respectively capture distribution shifts in transition functions and observation. The vector $\boldsymbol{\theta}_r = (\boldsymbol{\theta}_r^{\mathrm{dyn}}, \boldsymbol{\theta}_r^{\mathrm{obs}})$ has a constant value in each domain but varies across domains. The formulation of the nonstationary time-series model is in line with [30]. The nonstationary model is used to establish the identifiability results under nonstationary cases in Section 4.2, where we show that the violation of stationarity in both ways can even further improve the identifiability results. We first present the two nonstationary cases. **(1) Changing Causal Dynamics**. The causal influences between the latent temporal processes are changed across domains in this setting. We model it by adding the transition change factors $\boldsymbol{\theta}_r^{\mathrm{dyn}}$ as inputs to the transition function: $z_{it} = f_i\left(\{z_{j,t-\tau}|z_{j,t-\tau} \in \mathbf{Pa}(z_{it})\}, \boldsymbol{\theta}_r^{\mathrm{dyn}}, \epsilon_{it}\right)$. **(2) Global Observation Changes.** The global properties of the time series (e.g., video styles) are changed across domains in this setting. Our model captures them using latent variables that represent global styles; these latent variables are generated by a bijection $f_i$ that transforms the noise terms $\epsilon_{i,t}$ into the latent with change factor $\boldsymbol{\theta}_r^{\mathrm{obs}}$: $z_{i,t} = f_i\left(\boldsymbol{\theta}_r^{\mathrm{obs}}, \epsilon_{i,t}\right)$. Finally, we can deal with a more general nonstationary case by combining the three types of latent processes in the latent space in a modular way. **(3) Modular Distribution Shifts.** The latent space has three blocks $\mathbf{z}_t = (\mathbf{z}_t^{\mathrm{fix}}, \mathbf{z}_t^{\mathrm{chg}}, \mathbf{z}_t^{\mathrm{obs}})$ where $z_{s,t}^{\mathrm{fix}}$ is the $s^{\mathrm{th}}$ component of the fixed dynamics parts, $z_{c,t}^{\mathrm{chg}}$ is the $c^{\mathrm{th}}$ component of the changing dynamics parts, and $\mathbf{z}_{o,t}^{\mathrm{obs}}$ is the $o^{\mathrm{th}}$ component of the observation changes. The functions $[f_s, f_c, f_o]$

$$
\left\{
\begin{array}{ll}
z_{s,t}^{\mathrm{fix}} & = f_s\left(\{z_{i,t-\tau}|z_{i,t-\tau} \in \mathbf{Pa}(z_{s,t}^{\mathrm{fix}})\}, \epsilon_{s,t}\right), \\
z_{c,t}^{\mathrm{chg}} & = f_c\left(\{z_{i,t-\tau}|z_{i,t-\tau} \in \mathbf{Pa}(z_{c,t}^{\mathrm{chg}})\}, \boldsymbol{\theta}_r^{\mathrm{dyn}}, \epsilon_{c,t}\right), \\
z_{o,t}^{\mathrm{obs}} & = f_o\left(\boldsymbol{\theta}_r^{\mathrm{obs}}, \epsilon_{o,t}\right), \\
\mathbf{x}_t & = \mathbf{g}(\mathbf{z}_t).
\end{array}
\right.
\tag{1}
$$

capture fixed and changing transitions and observation changes for each dimension of $\mathbf{z}_t$ in Eq. 1.

### 3.2 Identifiability of Latent Causal Processes and Time-Delayed Latent Causal Relations

We define the identifiability of time-delayed latent causal processes in the representation function space in **Definition 1**. Furthermore, if the estimated latent processes can be identified at least up to permutation and component-wise invertible nonlinearities, the latent causal relations are also immediately identifiable because conditional independence relations fully characterize time-delayed causal relations in a time-delayed causally sufficient system, in which there are no latent causal confounders in the (latent) causal processes. Note that invertible component-wise transformations on latent causal processes do not change their conditional independence relations.

**Definition 1** (Identifiable Latent Causal Processes). *Formally let $\{\mathbf{x}_t\}_{t=1}^T$ be a sequence of observed variables generated by the true temporally causal latent processes specified by $(f_i, \boldsymbol{\theta}_r, p(\epsilon_i), \mathbf{g})$ given in Eq. 1. A learned generative model $(\hat{f}_i, \hat{\boldsymbol{\theta}}_r, \hat{p}(\epsilon_i), \hat{\mathbf{g}})$ is observationally equivalent to $(f_i, \boldsymbol{\theta}_r, p(\epsilon_i), \mathbf{g})$ if the model distribution $p_{\hat{f}, \hat{\boldsymbol{\theta}}_r, \hat{p}_{\epsilon}, \hat{\mathbf{g}}}(\{\mathbf{x}_t\}_{t=1}^T)$ matches the data distribution $p_{f, \boldsymbol{\theta}_r, p_{\epsilon}, \mathbf{g}}(\{\mathbf{x}_t\}_{t=1}^T)$ everywhere. We say latent causal processes are identifiable if observational equivalence can lead to identifiability of the latent variables up to permutation $\pi$ and component-wise invertible transformation $T$:*

$$
p_{\hat{f}_i, \hat{\boldsymbol{\theta}}_r, \hat{p}_{\epsilon_i}, \hat{\mathbf{g}}}(\{\mathbf{x}_t\}_{t=1}^T) = p_{f_i, \boldsymbol{\theta}_r, p_{\epsilon_i}, \mathbf{g}}(\{\mathbf{x}_t\}_{t=1}^T) \Rightarrow \hat{\mathbf{g}}(\mathbf{x}_t) = \mathbf{g} \circ \pi \circ T, \quad \forall \mathbf{x}_t \in \mathcal{X},
\tag{2}
$$

*where $\mathcal{X}$ is the observation space.*

## 4 Identifiability Theory

We establish the identifiability theory of nonparametric time-delayed latent causal processes under three different types of distribution shifts. W.l.o.g., we consider the latent processes with maximum time lag $L = 1$. The extentions to arbitrary time lags are discussed in Appendix S1.5. Let $k$ be the element index of the latent space $\mathbf{z}_t$ and the latent size is $n$. In particular, **(1)** under fixed temporal causal influences, we leverage the distribution changes $p(z_{k,t}|\mathbf{z}_{t-1})$ for different values

of $\mathbf{z}_{t-1}$; **(2)** when the underlying causal relations change over time, we exploit the changing causal influences on $p(z_{k,t}|\mathbf{z}_{t-1}, u_r)$ under different domain $u_r$, and **(3)** under global observation changes, the nonstationarity $p(z_{k,t}|u_r)$ under different values of $u_r$ is exploited. The proofs are provided in Appendix S1. The comparisons between existing theories are in Appendix S1.3.

## 4.1 Identifiability under Fixed Temporal Causal Influence

Let $\eta_{kt} \triangleq \log p(z_{k,t}|\mathbf{z}_{t-1})$. Assume that $\eta_{kt}$ is twice differentiable in $z_{k,t}$ and is differentiable in $z_{l,t-1}$, $l = 1, 2, ..., n$. Note that the parents of $z_{k,t}$ may be only a subset of $\mathbf{z}_{t-1}$; if $z_{l,t-1}$ is not a parent of $z_{k,t}$, then $\frac{\partial \eta_{kt}}{\partial z_{l,t-1}} = 0$. Below we provide a *sufficient condition* for the identifiability of $\mathbf{z}_t$, followed by a discussion of specific unidentifiable and identifiable cases to illustrate how general it is.

**Theorem 1** (Identifiablity under a Fixed Temporal Causal Model). *Suppose there exists invertible function $\hat{\mathbf{g}}$ that maps $\mathbf{x}_t$ to $\hat{\mathbf{z}}_t$, i.e.,*

$$\hat{\mathbf{z}}_t = \hat{\mathbf{g}}(\mathbf{x}_t) \tag{3}$$

*such that the components of $\hat{\mathbf{z}}_t$ are mutually independent conditional on $\hat{\mathbf{z}}_{t-1}$. Let*

$$\mathbf{v}_{k,t} \triangleq \left( \frac{\partial^2 \eta_{kt}}{\partial z_{k,t} \partial z_{1,t-1}}, \frac{\partial^2 \eta_{kt}}{\partial z_{k,t} \partial z_{2,t-1}}, ..., \frac{\partial^2 \eta_{kt}}{\partial z_{k,t} \partial z_{n,t-1}} \right)^{\mathsf{T}}, \mathring{\mathbf{v}}_{k,t} \triangleq \left( \frac{\partial^3 \eta_{kt}}{\partial z_{k,t}^2 \partial z_{1,t-1}}, \frac{\partial^3 \eta_{kt}}{\partial z_{k,t}^2 \partial z_{2,t-1}}, ..., \frac{\partial^3 \eta_{kt}}{\partial z_{k,t}^2 \partial z_{n,t-1}} \right)^{\mathsf{T}}. \tag{4}$$

*If for each value of $\mathbf{z}_t$, $\mathbf{v}_{1,t}, \mathring{\mathbf{v}}_{1,t}, \mathbf{v}_{2,t}, \mathring{\mathbf{v}}_{2,t}, ..., \mathbf{v}_{n,t}, \mathring{\mathbf{v}}_{n,t}$, as $2n$ vector functions in $z_{1,t-1}, z_{2,t-1}, ..., z_{n,t-1}$, are linearly independent, then $\mathbf{z}_t$ must be an invertible, component-wise transformation of a permuted version of $\hat{\mathbf{z}}_t$.*

The linear independence condition in Theorem 1 is the core condition to guarantee the identifiability of $\mathbf{z}_t$ from the observed $\mathbf{x}_t$. To make this condition more intuitive, below we consider specific unidentifiable cases, in which there is no temporal dependence in $\mathbf{z}_t$ or the noise terms in $\mathbf{z}_t$ are additive Gaussian, and two identifiable cases, in which $\mathbf{z}_t$ has additive, heterogeneous noise or follows some linear, non-Gaussian temporal process.

Let us start with two unidentifiable cases. In case N1, $\mathbf{t}_t$ is an independent and identically distributed (i.i.d.) process, i.e., there is no causal influence from any component of $\mathbf{z}_{t-1}$ to any $z_{k,t}$. In this case, $\mathbf{v}_{k,t}$ and $\mathring{\mathbf{v}}_{k,t}$ (defined in Eq. 4) are always $\mathbf{0}$ for $k = 1, 2, ..., n$, since $p(z_{k,t}|\mathbf{z}_{t-1})$ does not involve $\mathbf{z}_{t-1}$. So the linear independence condition is violated. In fact, this is the regular nonlinear ICA problem with i.i.d. data, and it is well-known that the underlying independent variables are not identifiable [5]. In case $N_2$, all $z_{k,t}$ follow an additive noise model with Gaussian noise terms, i.e.,

$$\mathbf{z}_t = \mathbf{q}(\mathbf{z}_{t-1}) + \epsilon_t, \tag{5}$$

where $\mathbf{q}$ is a transformation and the components of the Gaussian vector $\epsilon_t$ are independent and also independent from $\mathbf{z}_{t-1}$. Then $\frac{\partial^2 \eta_{kt}}{\partial z_{k,t}^2}$ is constant, and $\frac{\partial^3 \eta_{kt}}{\partial z_{k,t}^2 \partial z_{l,t-1}} \equiv 0$, violating the linear independence condition. In the following proposition we give some alternative solutions and verify the unidentifiability in this case.

**Proposition 1** (Unidentifiability under Gaussian Noise). *Suppose $\mathbf{x}_t = \mathbf{g}(\mathbf{z}_t)$ was generated by Eq. 5, where the components of $\epsilon_t$ are mutually independent Gaussian and also independent from $\mathbf{z}_{t-1}$. Then any $\hat{\mathbf{z}}_t = \mathbf{D}_1 \mathbf{U} \mathbf{D}_2 \cdot \mathbf{z}_t$, where $\mathbf{D}_1$ is an arbitrary non-singular diagonal matrix, $\mathbf{U}$ is an arbitrary orthogonal matrix, and $\mathbf{D}_2$ is a diagonal matrix with $\mathbb{V}ar^{-1/2}(\epsilon_{k,t})$ as its $k^{th}$ diagonal entry, is a valid solution to satisfy the condition that the components of $\hat{\mathbf{z}}_t$ are mutually independent conditional on $\hat{\mathbf{z}}_{t-1}$.*

Roughly speaking, for a randomly chosen conditional density function $p(z_{k,t}|\mathbf{z}_{t-1})$ in which $z_{k,t}$ is not independent from $\mathbf{z}_{t-1}$ (i.e., there is temporal dependence in the latent processes) and which does not follow an additive noise model with Gaussian noise, the chance for its specific second- and third-order partial derivatives to be linearly dependent is slim. Now let us consider two cases in which the latent temporally processes $\mathbf{z}_t$ are naturally identifiable. First, consider case Y1, where $z_{k,t}$ follows a heterogeneous noise process, in which the noise variance depends on its parents:

$$z_{k,t} = q_k(\mathbf{z}_{t-1}) + \frac{1}{b_k(\mathbf{z}_{t-1})} \epsilon_{k,t}. \tag{6}$$

Here we assume $\epsilon_{k,t}$ is standard Gaussian and $\epsilon_{1,t}, \epsilon_{2,t}, .., \epsilon_{n,t}$ are mutually independent and independent from $\mathbf{z}_{t-1}$. $\frac{1}{b_k}$, which depends on $\mathbf{z}_{t-1}$, is the standard deviation of the noise in $z_{k,t}$. (For

conciseness, we drop the argument of $b_k$ and $q_k$ when there is no confusion.) Note that in this model, if $q_k$ is 0 for all $k = 1, 2, ..., n$, it reduces to a multiplicative noise model. The identifiability result of $\mathbf{z}_t$ is established in the following corollary.

**Corollary 1** (Identifiability under Heterogeneous Noise). *Suppose* $\mathbf{x}_t = \mathbf{g}(\mathbf{z}_t)$ *was generated according to Eq. 6, and Eq. 3 holds true. If* $b_k \cdot \frac{\partial b_k}{\partial \mathbf{z}_{t-1}}$ *and* $b_k \cdot \frac{\partial b_k}{\partial \mathbf{z}_{t-1}}(z_{k,t} - q_k) - b_k^2 \cdot \frac{\partial q_k}{\partial \mathbf{z}_{t-1}}$, *with* $k = 1, 2, ..., n$, *which are in total $2n$ function vectors in* $\mathbf{z}_{t-1}$, *are linearly independent, then* $\mathbf{z}_t$ *must be an invertible, component-wise transformation of a permuted version of* $\hat{\mathbf{z}}_t$.

Let us then consider another special case, denoted by Y2, with a linear, non-Gaussian temporal model for $\mathbf{z}_t$: the latent processes follow Eq. 5, with $\mathbf{q}$ being a linear transformation and $\epsilon_{k,t}$ following a particular class of non-Gaussian distributions. The following corollary shows that $\mathbf{z}_t$ is identifiable as long as each $z_{k,t}$ receives causal influences from some components of $\mathbf{z}_{t-1}$.

**Corollary 2** (Identifiability under a Specific Linear, Non-Gaussian Model for Latent Processes). *Suppose* $\mathbf{x}_t = \mathbf{g}(\mathbf{z}_t)$ *was generated according to Eq. 5, in which* $\mathbf{q}$ *is a linear transformation and for each* $z_{k,t}$, *there exists at least one* $k'$ *such that* $c_{k,k'} \triangleq \frac{\partial z_{k,t}}{\partial z_{k',t-1}} \neq 0$. *Assume the noise term* $\epsilon_{k,t}$ *follows a zero-mean generalized normal distribution:*

$$p(\epsilon_{k,t}) \propto e^{-\lambda|\epsilon_{k,t}|^\beta}, \text{ with positive } \lambda \text{ and } \beta > 2 \text{ and } \beta \neq 3. \tag{7}$$

*If Eq. 3 holds, then* $\mathbf{z}_t$ *must be an invertible, component-wise transformation of permuted* $\hat{\mathbf{z}}_t$.

## 4.2 Further Benefits from Changing Causal Influences

Let $\eta_{kt}(u_r) \triangleq \log p(z_{k,t}|\mathbf{z}_{t-1}, u_r)$ where $r = 1, ..., m$. LEAP [14] established the identifiability of the latent temporal causal processes $\mathbf{z}_t$ in certain nonstationary cases, under the condition that the noise term in each $z_{k,t}$, relative to its parents in $\mathbf{z}_{t-1}$, changes across $m$ contexts corresponding to $\mathbf{u} = u_1, u_2, ..., u_m$. Here we show that the identifiability result shown in the previous section can further benefit from nonstationarity of the causal model, and that our identifiability condition is generally much weaker than that in [14]: we allow changes in the noise term or causal influence on $z_{k,t}$ from its parents in $\mathbf{z}_{t-1}$, and our "sufficient variability" condition is just a necessary condition for that in [14] because of the additional information that one can leverage. Let $\mathbf{v}_{k,t}(u_r)$ be $\mathbf{v}_{k,t}$, which is defined in Eq. 4, in the $u_r$ context. Similarly, Let $\mathring{\mathbf{v}}_{k,t}(u_r)$ be $\mathring{\mathbf{v}}_{k,t}$ in the $u_r$ context. Let

$$\mathbf{s}_{k,t} \triangleq \left(\mathbf{v}_{k,t}(u_1)^\intercal, ..., \mathbf{v}_{k,t}(u_m)^\intercal, \Delta_2^2, ..., \Delta_m^2\right)^\intercal, \mathring{\mathbf{s}}_{k,t} \triangleq \left(\mathring{\mathbf{v}}_{k,t}(u_1)^\intercal, ..., \mathring{\mathbf{v}}_{k,t}(u_m)^\intercal, \Delta_2, ..., \Delta_m\right)^\intercal, \tag{8}$$

where $\Delta_r^2 = \frac{\partial^2 \eta_{kt}(u_r)}{\partial z_{k,t}^2} - \frac{\partial^2 \eta_{kt}(u_{r-1})}{\partial z_{k,t}^2}$ and $\Delta_r = \frac{\partial \eta_{kt}(u_r)}{\partial z_{k,t}} - \frac{\partial \eta_{kt}(u_{r-1})}{\partial z_{k,t}}$. As provided below, in our case, the identifiablity of $\mathbf{z}_t$ is guaranteed by the linear independence of the whole function vectors $\mathbf{s}_{k,t}$ and $\mathring{\mathbf{s}}_{k,t}$, with $k = 1, 2, ..., n$. However, the identifiability result in [14] relies on the linear independence of only the last $m - 1$ components of $\mathbf{s}_{k,t}$ and $\mathring{\mathbf{s}}_{k,t}$ with $k = 1, 2, ..., n$; this linear independence is generally a much stronger and restricted condition.

**Theorem 2** (Identifiability under Changing Causal Dynamics). *Suppose* $\mathbf{x}_t = \mathbf{g}(\mathbf{z}_t)$ *and that the conditional distribution* $p(z_{k,t} | \mathbf{z}_{t-1})$ *may change across $m$ values of the context variable* $\mathbf{u}$, *denoted by* $u_1, u_2, ..., u_m$. *Suppose the components of* $\mathbf{z}_t$ *are mutually independent conditional on* $\mathbf{z}_{t-1}$ *in each context. Assume that the components of* $\hat{\mathbf{z}}_t$ *produced by Eq. 3 are also mutually independent conditional on* $\hat{\mathbf{z}}_{t-1}$. *If the $2n$ function vectors* $\mathbf{s}_{k,t}$ *and* $\mathring{\mathbf{s}}_{k,t}$, *with* $k = 1, 2, ..., n$, *are linearly independent, then* $\hat{\mathbf{z}}_t$ *is a permuted invertible component-wise transformation of* $\mathbf{z}_t$.

**Theorem 3** (Identifiability under Observation Changes). *Suppose* $\mathbf{x}_t = \mathbf{g}(\mathbf{z}_t)$ *and that the conditional distribution* $p(z_{k,t} | \mathbf{u})$ *may change across $m$ values of the context variable* $\mathbf{u}$, *denoted by* $u_1, u_2, ..., u_m$. *Suppose the components of* $\mathbf{z}_t$ *are mutually independent conditional on* $\mathbf{u}$ *in each context. Assume that the components of* $\hat{\mathbf{z}}_t$ *produced by Eq. 3 are also mutually independent conditional on* $\hat{\mathbf{z}}_{t-1}$. *If the $2n$ function vectors* $\mathbf{s}_{k,t}$ *and* $\mathring{\mathbf{s}}_{k,t}$, *with* $k = 1, 2, ..., n$, *are linearly independent, then* $\hat{\mathbf{z}}_t$ *is a permuted invertible component-wise transformation of* $\mathbf{z}_t$.

**Corollary 3** (Identifiability under Modular Distribution Shifts). *Assume the data generating process in Eq. 1. If the three partitioned latent components* $\mathbf{z}_t = (\mathbf{z}_t^{fix}, \mathbf{z}_t^{chg}, \mathbf{z}_t^{obs})$ *respectively satisfy the conditions in* **Theorem 1**, **Theorem 2**, *and* **Theorem 3**, *then* $\mathbf{z}_t$ *must be an invertible, component-wise transformation of a permuted version of* $\hat{\mathbf{z}}_t$.

# 5 Our Approach

## 5.1 `TDRL` : Temporally Disentangled Representation Learning

Given our identifiability results, we propose `TDRL` framework to estimate the latent causal dynamics under modular distribution shifts, by extending Sequential Variational Auto-Encoders [31] with tailored modules to model different distribution shifts, and enforcing the conditions in Sec. 4 as constraints. We give the estimation procedure of the latent causal dynamics model in Eq. 1. The model architecture is showcased in Fig. 2. The framework has the following three major components. The implementation details are in Appendix S3.3. Specifically, we leverage the partitioned estimated latent subspaces $\hat{\mathbf{z}}_t = (\hat{\mathbf{z}}_t^{\text{fix}}, \hat{\mathbf{z}}_t^{\text{chg}}, \hat{\mathbf{z}}_t^{\text{obs}})$ and model their distribution changes in conditional transition priors. We use $[f_s, f_c, f_o]$ to capture causal relations as in Eq. 1, where $f_s$ captures fixed causal influences, $f_c$ for changing causal influences, and $f_o$ for observation changes. Accordingly, we learn $[\hat{f}_s^{-1}, \hat{f}_c^{-1}, \hat{f}_o^{-1}]$ to output random process noise from the estimated direct cause (lagged states $\hat{\mathbf{z}}_{\text{Hx}}$) and effect (current states) variables $\hat{\mathbf{z}}_t$.

**Change Factor Representation** We learn to embed domain index $\mathbf{u}_r$ into low-dimensional change factors $(\hat{\boldsymbol{\theta}}_r^{\text{dyn}}, \hat{\boldsymbol{\theta}}_r^{\text{obs}})$ in Fig. 2 and insert them as external inputs to the (inverse) dynamics function $\hat{f}_c^{-1}(\hat{\boldsymbol{\theta}}_r^{\text{dyn}})$, or the observation bijector $\hat{f}_o^{-1}(\hat{\boldsymbol{\theta}}_r^{\text{obs}})$, respectively. And hence the distribution shifts are captured and utilized in the implementation.

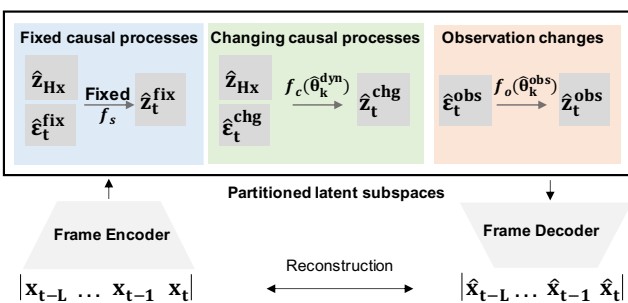

Figure 2: `TDRL` describes each domain with change factors $(\hat{\boldsymbol{\theta}}_r^{\text{dyn}}, \hat{\boldsymbol{\theta}}_r^{\text{obs}})$ and inserts them into the prior models of the partitioned latent processes. The posteriors of the latent variables are inferred from image frames with variational autoencoder.

**Modular Prior Network** We follow standard conditional normalizing flow formulation [32, 14]. Let $\mathbf{z}_{\text{Hx}}$ denote the lagged latent variables up to maximum time lag $L$. In particular, for **1) fixed causal dynamics processes** $\hat{z}_t^{\text{fix}}$, their transition priors are obtained by first learning inverse transition functions $f_s^{-1}$ that take the estimated latent variables and output random noise terms, and applying the change of variables formula to the transformation: $p(\hat{z}_{s,t}^{\text{fix}}|\hat{\mathbf{z}}_{\text{Hx}}) = p_{\epsilon_s}\left(\hat{f}_s^{-1}(\hat{z}_{s,t}^{\text{fix}}, \hat{\mathbf{z}}_{\text{Hx}})\right)\left|\frac{\partial \hat{f}_s^{-1}}{\partial \hat{z}_{s,t}^{\text{fix}}}\right|$; for **2) changing causal dynamics**, we evaluate $p(\hat{z}_{c,t}^{\text{chg}}|\hat{\mathbf{z}}_{\text{Hx}}, \mathbf{u}_r) = p_{\epsilon_c}\left(\hat{f}_c^{-1}(\hat{z}_{c,t}^{\text{chg}}, \hat{\mathbf{z}}_{\text{Hx}}, \hat{\boldsymbol{\theta}}_r^{\text{dyn}})\right)\left|\frac{\partial \hat{f}_c^{-1}}{\partial \hat{z}_{c,t}^{\text{chg}}}\right|$ by learning a holistic inverse dynamics $\hat{f}_c^{-1}$ that takes the estimated change factors for dynamics $\hat{\boldsymbol{\theta}}_r^{\text{dyn}}$ as inputs, and similarly for **3) global observation changes** $\hat{z}_t^{\text{obs}}$, we learn to project them to invariant noise terms by $\hat{f}_o^{-1}$ which takes the change factors $\boldsymbol{\theta}_r^{\text{obs}}$ as arguments, and obtains $p(\hat{z}_{o,t}^{\text{obs}}|\mathbf{u}_r) = p_{\epsilon_o}\left(\hat{f}_o^{-1}(\hat{z}_{o,t}^{\text{obs}}, \hat{\boldsymbol{\theta}}_r^{\text{obs}})\right)\left|\frac{\partial \hat{f}_o^{-1}}{\partial \hat{z}_{o,t}^{\text{obs}}}\right|$ as the prior.

**Conditional independence of the estimated latent variables** $p(\hat{z}_t|\hat{\mathbf{z}}_{\text{Hx}})$ is enforced by summing up all estimated component densities when obtaining the joint $p(\mathbf{z}_t|\mathbf{z}_{\text{Hx}}, \mathbf{u})$ in Eq. 9. Given that the Jacobian is lower-triangular, we can compute its determinant as the product of diagonal terms. The detailed derivations are given in Appendix S3.1.

$$\log p\left(\hat{\mathbf{z}}_t|\hat{\mathbf{z}}_{\text{Hx}}, \mathbf{u}_r\right) = \underbrace{\sum_{i=1}^{n} \log p(\hat{\epsilon}_i|\mathbf{u}_r)}_{\text{Conditional indepednce}} + \underbrace{\sum_{i=1}^{n} \log\left|\frac{\partial \hat{f}_i^{-1}}{\partial \hat{z}_{it}}\right|}_{\text{Lower-triangular Jacobian}} \tag{9}$$

**Factorized Inference** We infer the posteriors of each time step $q(\hat{\mathbf{z}}_t|\mathbf{x}_t)$ using only the observation at that time step, because in Eq. 1, $\mathbf{x}_t$ preserves all the information of the current system states so the joint probability $q(\hat{\mathbf{z}}_{1:T}|\mathbf{x}_{1:T})$ can be factorized into product of these terms. We approximate the posterior $q(\hat{\mathbf{z}}_t|\mathbf{x}_t)$ with an isotropic Gaussian with mean and variance from the inference network.

**Optimization** We train `TDRL` using the ELBO objective $\mathcal{L}_{\text{ELBO}} = \frac{1}{N} \sum_{i \in N} \mathcal{L}_{\text{Recon}} - \beta \mathcal{L}_{\text{KLD}}$, in which we use mean-squared error (MSE) for the reconstruction likelihood $\mathcal{L}_{\text{Recon}}$. The KL divergence

$\mathcal{L}_{\text{KLD}}$ is estimated via a sampling approach since with a learned nonparametric modular transition prior, the distribution does not have an explicit form. Specifically, we obtain the log-likelihood of the posterior, evaluate the prior $\log p\left(\hat{\mathbf{z}}_t|\hat{\mathbf{z}}_{\text{Hx}}, \mathbf{u}_r\right)$ in Eq. 9, and compute their mean difference in the dataset as the KL loss: $\mathcal{L}_{\text{KLD}} = \mathbb{E}_{\hat{\mathbf{z}}_t \sim q\left(\hat{\mathbf{z}}_t^{(i)}|\mathbf{x}_t^{(i)}\right)} \log q(\hat{\mathbf{z}}_t|\mathbf{x}_t) - \log p(\hat{\mathbf{z}}_t|\hat{\mathbf{z}}_{\text{Hx}}, \mathbf{u}_r)$.

## 5.2 Causal Visualization

For visualization purposes, when the underlying latent processes have sparse causal relations, we fit LassoNet [33] on the latent processes recovered by **TDRL** to interpret the causal relations. Specifically, we fit LassoNet to predict $\hat{\mathbf{z}}_t$ using the estimated history information $\hat{\mathbf{z}}_{\text{Hx}} = \{\hat{\mathbf{z}}_{t-\tau}\}_{\tau=1}^{L}$ up to maximum time lag $L$. Note that this postprocessing step is optional – the latent causal relations have already been captured in the learned transition functions in **TDRL** . Also, our identifiability conditions do not rely on the sparsity of causal relations in the latent processes.

# 6 Experiments

We evaluate the identifiability results of **TDRL** on a number of simulated and real-world time-series datasets. We first introduce the evaluation metrics and baselines. **(1) Evaluation Metrics.** To evaluate the identifiability of the latent variables, we compute Mean Correlation Coefficient (MCC) on the test dataset. MCC is a standard metric in the ICA literature for continuous variables which measure the identifiability of the learned latent causal processes. MCC is close to 1 when latent variables are identifiable up to permutation and component-wise invertible transformation in the noiseless case. **(2) Baselines.** Nonlinear ICA methods are used: **(1)** BetaVAE [34] which ignores both history and nonstationarity information; **(2)** iVAE [9] and TCL [6] which leverage nonstationarity to establish identifiability but assumes independent factors, and **(3)** SlowVAE [12] and PCL [7] which exploit temporal constraints but assume independent sources and stationary processes, and **(4)** LEAP [14] which assumes nonstationary, causal processes but only models nonstationary noise. Two other disentangled deep state-space models with nonlinear dynamics models: Kalman VAE (KVAE [35]) and Deep Variational Bayes Filters (DVBF [36]), are also used for comparisons.

Table 2: MCC scores and their standard deviations for the three simulation settings over 3 random seeds. Note: The symbol "–" represents that this method is not applicable to this dataset.

| Experiment Settings | Method | | | | | | | | |
|---|---|---|---|---|---|---|---|---|---|
| | TDRL | LEAP | SlowVAE | PCL | i-VAE | TCL | betaVAE | KVAE | DVBF |
| Fixed | **0.954 ±0.009** | – | 0.411 ±0.022 | 0.516 ±0.043 | – | – | 0.353 ±0.001 | 0.832 ±0.038 | 0.778 ±0.045 |
| Changing | **0.958 ±0.017** | 0.726 ±0.187 | 0.511 ±0.062 | 0.599 ±0.041 | 0.581 ±0.083 | 0.399 ±0.021 | 0.523 ±0.009 | 0.711 ±0.062 | 0.648 ±0.071 |
| Modular | **0.993 ±0.001** | 0.657 ±0.108 | 0.406 ±0.045 | 0.564 ±0.049 | 0.557 ±0.005 | 0.297 ±0.078 | 0.433 ±0.045 | 0.632 ±0.048 | 0.678 ±0.074 |

## 6.1 Simulated Experiments

We generate synthetic datasets that satisfy our identifiability conditions in the theorems following the procedures in Appendix S2.1.1. As in Table 2, our framework can recover the latent processes under fixed dynamics (heterogeneous noise model), under changing causal dynamics, and under modular distribution shifts with high MCCs (>0.95). The baselines that do not exploit history (i.e., $\beta$VAE, i-VAE, TCL), with independent source assumptions (SlowVAE, PCL), considers limited nonstationary cases (LEAP) distort the identifiability results. KVAE and DVBF achieve MCCs (0.8) under fixed dynamics but distorts the identifiability under changing dynamics and modular shift setting because they don't model changing causal relations and global observation changes.

## 6.2 Real-world Applications

**Video Data – Modified Cartpole Environment**    We evaluate **TDRL** on the modified cartpole [30] video dataset and compare the performances with the baselines. Modified Cartpole is a nonlinear dynamical system with cart positions $x_t$ and pole angles $\theta_t$ as the true state variables. The dataset descriptions are in Appendix S2.1.2. We use 6 source domains with different gravity values $g = \{5, 10, 15, 20, 25, 30\}$. Together with the 2 discrete actions (i.e., left and right), we have 12 segments of data with changing causal dynamics. We fit **TDRL** with two-dimensional change factors $\boldsymbol{\theta}_r^{\text{dyn}}$. We set the latent size $n = 8$ and the lag number $L = 2$. In Fig. 3, the latent causal processes are recovered, as seen from (a) high MCC for the latent causal processes; (b) the latent factors are estimated up to

component-wise transformation; (c) **TDRL** outperforms the baselines and (d) the latent traversals confirm the two recovered latent variables correspond to the position and pole angle.

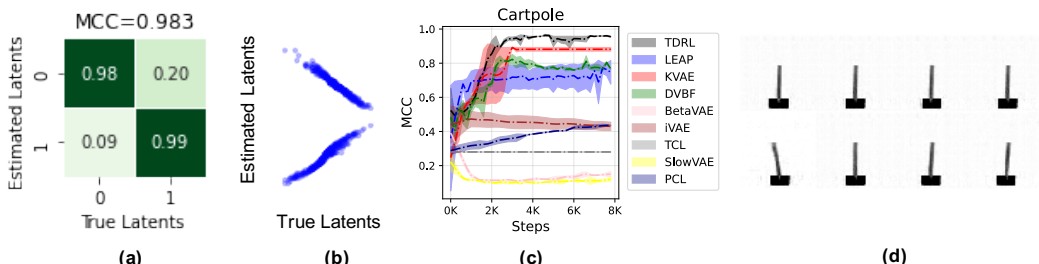

Figure 3: Modified Cartpole results: (a) MCC for causally-related factors; (b) scatterplots between estimated and true factors; (c) baseline comparisons, and (d) latent traversal on a fixed video frame;.

**Motion Capture Data – CMU-Mocap**    We experimented with another real-world motion capture dataset (CMU-Mocap). The dataset descriptions are in Appendix S2.1.2. We fit **TDRL** with 11 trials of motion capture data for subject #8. The 11 trials contain walk cycles with very different styles (e.g., slow walk, stride). We set latent size $n = 3$ and lag number $L = 2$. The differences between trials are captured through learning the 2-dimensional change factors for each trial. In Fig. 4(a), the learned change factors group similar walk styles into clusters; in Panel (c), three latent variables (which seem to be pitch, yaw, roll rotations) are found to explain most of the variances of human walk cycles. The learned latent coordinates (Panel b) show smooth cyclic patterns with differences among different walking styles. For the discovered skeleton (Panel d), roll and pitch of walking are found to be causally-related while yaw has independent dynamics.

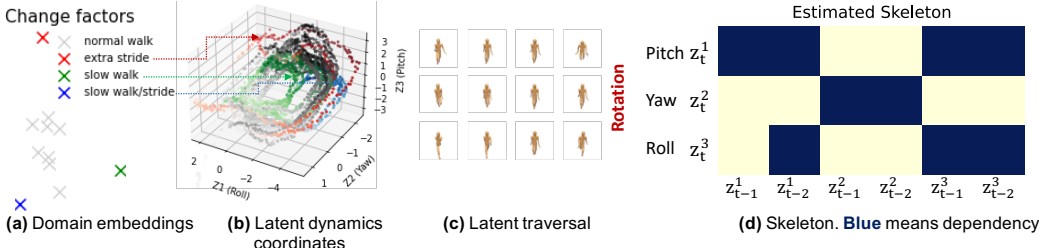

Figure 4: CMU-Mocap results (Subject #8): (a) learned change factors; (b) latent coordinates dynamics for 11 trials; (c) ) latent traversal by rendering the reconstructed point clouds into the video frame; (d) estimated causal skeleton (blue indicates dependency).

## 7    Conclusion

In this paper, without relying on parametric or distribution assumptions, we established identifiability theories for nonparametric latent causal processes from their observed nonlinear mixtures in stationary environments and under unknown distribution shifts. **The basic limitation** of this work is that the underlying latent processes are assumed to have no instantaneous causal relations but only time-delayed influences. If the time resolution of the observed time series is the same as the causal frequency (or even higher), this assumption naturally holds. However, if the resolution of the time series is much lower, then it is usually violated and one has to find a way to deal with instantaneous causal relations. On the other hand, it is worth mentioning that the assumptions are generally testable; if the assumptions are actually violated, one can see that the results produced by our method may not be reliable. Extending our theories and framework to address the issue of instantaneous dependency or instantaneous causal relations will be one line of our future work. In addition, Empirically exploration of the merits of the identified latent causal graph in terms of few-shot transfer to new environments [37], domain adaptation [38], forecasting [39, 40] and control [30] is also one important future step.

## Acknowledgment

Kun Zhang was partially supported by the National Institutes of Health (NIH) under Contract R01HL159805, by the NSF-Convergence Accelerator Track-D award #2134901, by a grant from Apple Inc., and by a grant from KDDI Research Inc.

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
