*Supplement to*

## "Temporally Disentangled Representation Learning"

Appendix organization:

---

---

## S1 Identifiability Theory

The observed variables were generated according to :

$$\mathbf{x}_t = \mathbf{g}(\mathbf{z}_t), \tag{10}$$

in which $\mathbf{g}$ is invertible, and $z_{it}$, as the $i$th component of $\mathbf{z}_t$, is generated by (some) components of $\mathbf{z}_{t-1}$ and noise $E_{it}$. $E_{1t}, E_{2t}, ..., E_{nt}$ are mutually independent. In other words, the components of $\mathbf{z}_t$ are mutually independent conditional on $\mathbf{z}_{t-1}$. Let $\eta_{kt} \triangleq \log p(z_{kt}|\mathbf{z}_{t-1})$. Assume that $\eta_k(t)$ is twice differentiable in $z_{kt}$ and is differentiable in $z_{l,t-1}$, $l = 1, 2, ..., n$. Note that the parents of $z_{kt}$ may be only a subset of $\mathbf{z}_{t-1}$; if $z_{l,t-1}$ is not a parent of $z_{kt}$, then $\frac{\partial \eta_k}{\partial z_{l,t-1}} = 0$.

### S1.1 Proof for Theorem 1

**Theorem S1** (Identifiablity under a Fixed Temporal Causal Model)**.** *Suppose there exists invertible function $\mathbf{f}$, which is the estimated mixing function (i.e., we use $\mathbf{f}$ and $\hat{\mathbf{g}}$ interchangeably in Appendix)*

*that maps $\mathbf{x}_t$ to $\hat{\mathbf{z}}_t$, i.e.,*

$$\hat{\mathbf{z}}_t = \mathbf{f}(\mathbf{x}_t) \tag{11}$$

*such that the components of $\hat{\mathbf{z}}_t$ are mutually independent conditional on $\hat{\mathbf{z}}_{t-1}$. Let*

$$\mathbf{v}_{k,t} \triangleq \Big( \frac{\partial^2 \eta_{kt}}{\partial z_{k,t} \partial z_{1,t-1}}, \frac{\partial^2 \eta_{kt}}{\partial z_{k,t} \partial z_{2,t-1}}, ..., \frac{\partial^2 \eta_{kt}}{\partial z_{k,t} \partial z_{n,t-1}} \Big)^{\mathsf{T}}$$

$$\mathring{\mathbf{v}}_{k,t} \triangleq \Big( \frac{\partial^3 \eta_{kt}}{\partial z_{k,t}^2 \partial z_{1,t-1}}, \frac{\partial^3 \eta_{kt}}{\partial z_{k,t}^2 \partial z_{2,t-1}}, ..., \frac{\partial^3 \eta_{kt}}{\partial z_{k,t}^2 \partial z_{n,t-1}} \Big)^{\mathsf{T}}. \tag{12}$$

*If for each value of $\mathbf{z}_t$, $\mathbf{v}_{1t}, \mathring{\mathbf{v}}_{1t}, \mathbf{v}_{2t}, \mathring{\mathbf{v}}_{2t}, ..., \mathbf{v}_{nt}, \mathring{\mathbf{v}}_{nt}$, as $2n$ vector functions in $z_{1,t-1}$, $z_{2,t-1}$, ..., $z_{n,t-1}$, are linearly independent, then $\mathbf{z}_t$ must be an invertible, component-wise transformation of a permuted version of $\hat{\mathbf{z}}_t$.*

*Proof.* Combining Eq. 10 and Eq. 11 gives $\mathbf{z}_t = \mathbf{g}^{-1}(\mathbf{f}^{-1}(\hat{\mathbf{z}}_t)) = \mathbf{h}(\hat{\mathbf{z}}_t)$, where $\mathbf{h} \triangleq \mathbf{g}^{-1} \circ \mathbf{f}^{-1}$. Since both $\mathbf{f}$ and $\mathbf{g}$ are invertible, $\mathbf{h}$ is invertible. Let $\mathbf{H}_t$ be the Jacobian matrix of the transformation $h(\hat{\mathbf{z}}_t)$, and denote by $\mathbf{H}_{kit}$ its $(k,i)$th entry.

First, it is straightforward to see that if the components of $\hat{\mathbf{z}}_t$ are mutually independent conditional on $\hat{\mathbf{z}}_{t-1}$, then for any $i \neq j$, $\hat{z}_{it}$ and $\hat{z}_{jt}$ are conditionally independent given $\hat{\mathbf{z}}_{t-1} \cup (\hat{\mathbf{z}}_t \setminus \{\hat{z}_{it}, \hat{z}_{jt}\})$. Mutual independence of the components of $\hat{\mathbf{z}}_t$ conditional on $\hat{\mathbf{z}}_{t-1}$ implies that $\hat{z}_{it}$ is independent from $\hat{\mathbf{z}}_t \setminus \{\hat{z}_{it}, \hat{z}_{jt}\}$ conditional on $\hat{\mathbf{z}}_{t-1}$, i.e.,

$$p(\hat{z}_{it} \,|\, \hat{\mathbf{z}}_{t-1}) = p(\hat{z}_{it} \,|\, \hat{\mathbf{z}}_{t-1} \cup (\hat{\mathbf{z}}_t \setminus \{\hat{z}_{it}, \hat{z}_{jt}\})).$$

At the same time, it also implies $\hat{z}_{it}$ is independent from $\hat{\mathbf{z}}_t \setminus \{\hat{z}_{it}\}$ conditional on $\hat{\mathbf{z}}_{t-1}$, i.e.,

$$p(\hat{z}_{it} \,|\, \hat{\mathbf{z}}_{t-1}) = p(\hat{z}_{it} \,|\, \hat{\mathbf{z}}_{t-1} \cup (\hat{\mathbf{z}}_t \setminus \{\hat{z}_{it}\})).$$

Combining the above two equations gives $p(\hat{z}_{it} \,|\, \hat{\mathbf{z}}_{t-1} \cup (\hat{\mathbf{z}}_t \setminus \{\hat{z}_{it}\})) = p(\hat{z}_{it} \,|\, \hat{\mathbf{z}}_{t-1} \cup (\hat{\mathbf{z}}_t \setminus \{\hat{z}_{it}, \hat{z}_{jt}\}))$, i.e., for $i \neq j$, $\hat{z}_{it}$ and $\hat{z}_{jt}$ are conditionally independent given $\hat{\mathbf{z}}_{t-1} \cup (\hat{\mathbf{z}}_t \setminus \{\hat{z}_{it}, \hat{z}_{jt}\})$.

We then make use of the fact that if $\hat{z}_{it}$ and $\hat{z}_{jt}$ are conditionally independent given $\hat{\mathbf{z}}_{t-1} \cup (\hat{\mathbf{z}}_t \setminus \{\hat{z}_{it}, \hat{z}_{jt}\})$, then

$$\frac{\partial^2 \log p(\hat{\mathbf{z}}_t, \hat{\mathbf{z}}_{t-1})}{\partial \hat{z}_{it} \partial \hat{z}_{jt}} = 0,$$

assuming the cross second-order derivative exists [41]. Since $p(\hat{\mathbf{z}}_t, \hat{\mathbf{z}}_{t-1}) = p(\hat{\mathbf{z}}_t \,|\, \hat{\mathbf{z}}_{t-1}) p(\hat{\mathbf{z}}_{t-1})$ while $p(\hat{\mathbf{z}}_{t-1})$ does not involve $\hat{z}_{it}$ or $\hat{z}_{jt}$, the above equality is equivalent to

$$\frac{\partial^2 \log p(\hat{\mathbf{z}}_t \,|\, \hat{\mathbf{z}}_{t-1})}{\partial \hat{z}_{it} \partial \hat{z}_{jt}} = 0. \tag{13}$$

The Jacobian matrix of the mapping from $(\mathbf{x}_{t-1}, \hat{\mathbf{z}}_t)$ to $(\mathbf{x}_{t-1}, \mathbf{z}_t)$ is $\begin{bmatrix} \mathbf{I} & \mathbf{0} \\ * & \mathbf{H}_t \end{bmatrix}$, where $*$ stands for a matrix, and the (absolute value of the) determinant of this Jacobian matrix is $|\mathbf{H}_t|$. Therefore $p(\hat{\mathbf{z}}_t, \mathbf{x}_{t-1}) = p(\mathbf{z}_t, \mathbf{x}_{t-1}) \cdot |\mathbf{H}_t|$. Dividing both sides of this equation by $p(\mathbf{x}_{t-1})$ gives

$$p(\hat{\mathbf{z}}_t \,|\, \mathbf{x}_{t-1}) = p(\mathbf{z}_t \,|\, \mathbf{x}_{t-1}) \cdot |\mathbf{H}_t|. \tag{14}$$

Since $p(\mathbf{z}_t \,|\, \mathbf{z}_{t-1}) = p(\mathbf{z}_t \,|\, \mathbf{g}(\mathbf{z}_{t-1})) = p(\mathbf{z}_t \,|\, \mathbf{x}_{t-1})$ and similarly $p(\hat{\mathbf{z}}_t \,|\, \hat{\mathbf{z}}_{t-1}) = p(\hat{\mathbf{z}}_t \,|\, \mathbf{x}_{t-1})$, Eq. 14 tells us

$$\log p(\hat{\mathbf{z}}_t \,|\, \hat{\mathbf{z}}_{t-1}) = \log p(\mathbf{z}_t \,|\, \mathbf{z}_{t-1}) + \log |\mathbf{H}_t| = \sum_{k=1}^{n} \eta_{kt} + \log |\mathbf{H}_t|. \tag{15}$$

Its partial derivative w.r.t. $\hat{z}_{it}$ is

$$\frac{\partial \log p(\hat{\mathbf{z}}_t \,|\, \hat{\mathbf{z}}_{t-1})}{\partial \hat{z}_{it}} = \sum_{k=1}^{n} \frac{\partial \eta_{kt}}{\partial z_{kt}} \cdot \frac{\partial z_{kt}}{\partial \hat{z}_{it}} - \frac{\partial \log |\mathbf{H}_t|}{\partial \hat{z}_{it}}$$

$$= \sum_{k=1}^{n} \frac{\partial \eta_{kt}}{\partial z_{kt}} \cdot \mathbf{H}_{kit} - \frac{\partial \log |\mathbf{H}_t|}{\partial \hat{z}_{it}}.$$

Its second-order cross derivative is

$$\frac{\partial^2 \log p(\hat{\mathbf{z}}_t \mid \hat{\mathbf{z}}_{t-1})}{\partial \hat{z}_{it} \partial \hat{z}_{jt}} = \sum_{k=1}^{n} \left( \frac{\partial^2 \eta_{kt}}{\partial z_{kt}^2} \cdot \mathbf{H}_{kit} \mathbf{H}_{kjt} + \frac{\partial \eta_{kt}}{\partial z_{kt}} \cdot \frac{\partial \mathbf{H}_{kit}}{\partial \hat{z}_{jt}} \right) - \frac{\partial^2 \log |\mathbf{H}_t|}{\partial \hat{z}_{it} \partial \hat{z}_{jt}}. \tag{16}$$

The above quantity is always 0 according to Eq. 13. Therefore, for each $l = 1, 2, ..., n$ and each value $z_{l,t-1}$, its partial derivative w.r.t. $z_{l,t-1}$ is always 0. That is,

$$\frac{\partial^3 \log p(\hat{\mathbf{z}}_t \mid \hat{\mathbf{z}}_{t-1})}{\partial \hat{z}_{it} \partial \hat{z}_{jt} \partial z_{l,t-1}} = \sum_{k=1}^{n} \left( \frac{\partial^3 \eta_{kt}}{\partial z_{kt}^2 \partial z_{l,t-1}} \cdot \mathbf{H}_{kit} \mathbf{H}_{kjt} + \frac{\partial^2 \eta_{kt}}{\partial z_{kt} \partial z_{l,t-1}} \cdot \frac{\partial \mathbf{H}_{kit}}{\partial \hat{z}_{jt}} \right) \equiv 0, \tag{17}$$

where we have made use of the fact that entries of $\mathbf{H}_t$ do not depend on $z_{l,t-1}$.

If for any value of $\mathbf{z}_t$, $\mathbf{v}_{1t}, \mathring{\mathbf{v}}_{1t}, \mathbf{v}_{2t}, \mathring{\mathbf{v}}_{2t}, ..., \mathbf{v}_{nt}, \mathring{\mathbf{v}}_{nt}$ are linearly independent, to make the above equation hold true, one has to set $\mathbf{H}_{kit} \mathbf{H}_{kjt} = 0$ or $i \neq j$. That is, in each row of $\mathbf{H}_t$ there is only one non-zero entry. Since $h$ is invertible, then $\mathbf{z}_t$ must be an invertible, component-wise transformation of a permuted version of $\hat{\mathbf{z}}_t$. $\qquad \square$

The linear independence condition in Theorem S1 is the core condition to guarantee the identifiability of $\mathbf{z}_t$ from the observed $\mathbf{x}_t$. Roughly speaking, for a randomly chosen conditional density function $p(z_{kt} \mid \mathbf{z}_{t-1})$, the chance for this constraint to hold on its second- and third-order partial derivatives is slim. For illustrative purposes, below we make this claim more precise, by considering a specific unidentifiable case, in which the noise terms in $\mathbf{z}_t$ are additive Gaussian, and two identifiable cases, in which $\mathbf{z}_t$ has additive, heterogeneous noise or follows some linear, non-Gaussian temporal process.

Let us start with an unidentifiable case. If all $z_{kt}$ follow the additive noise model with Gaussian noise terms, i.e.,

$$\mathbf{z}_t = \mathbf{q}(\mathbf{z}_{t-1}) + \mathbf{E}_t, \tag{18}$$

where $\mathbf{q}$ is a transformation and the components of the Gaussian vector $\mathbf{E}_t$ are independent and also independent from $\mathbf{z}_{t-1}$. Then $\frac{\partial^2 \eta_{kt}}{\partial z_{kt}^2}$ is constant, and $\frac{\partial^3 \eta_{kt}}{\partial z_{kt}^2 \partial z_{l,t-1}} \equiv 0$, violating the linear independence condition. In the following proposition we give some alternative solutions and verify the unidentifiability in this case.

**Proposition S1** (Unidentifiability under Gaussian noise). *Suppose $\mathbf{x}_t$ was generated according to Eq. 10 and Eq. 18, where the components of $\mathbf{E}_t$ are mutually independent Gaussian and also independent from $\mathbf{z}_{t-1}$. Then any $\hat{\mathbf{z}}_t = \mathbf{D}_1 \mathbf{U} \mathbf{D}_2 \cdot \mathbf{z}_t$, where $\mathbf{D}_1$ is an arbitrary non-singular diagonal matrix, $\mathbf{U}$ is an arbitrary orthogonal matrix, and $\mathbf{D}_2$ is a diagonal matrix with $\mathbb{V}ar^{-1/2}(E_{kt})$ as its kth diagonal entry, is a valid solution to satisfy the condition that the components of $\hat{\mathbf{z}}_t$ are mutually independent conditional on $\hat{\mathbf{z}}_{t-1}$.*

*Proof.* In this case we have

$$\hat{\mathbf{z}}_t = \mathbf{D}_1 \mathbf{U} \mathbf{D}_2 \cdot \mathbf{q}(\mathbf{z}_{t-1}) + \mathbf{D}_1 \mathbf{U} \mathbf{D}_2 \cdot \mathbf{E}_t.$$

It is easy to verify that the components of $\mathbf{D}_1 \mathbf{U} \mathbf{D}_2 \cdot \mathbf{E}_t$ are mutually independent and are independent from $\mathbf{D}_1 \mathbf{U} \mathbf{D}_2 \cdot \mathbf{q}(\mathbf{z}_{t-1})$. As a consequence, $\hat{\mathbf{z}}_t$ are mutually independent conditional on $\hat{\mathbf{z}}_{t-1}$. $\qquad \square$

Now let us consider some cases in which the latent temporally processes $\mathbf{z}_t$ are naturally identifiable under some technical conditions. Let us first consider the case where $z_{kt}$ follows a heterogeneous noise process, in which the noise variance depends on its parents:

$$a_{kt} = q_k(\mathbf{z}_{t-1}) + \frac{1}{b_k(\mathbf{z}_{t-1})} E_{kt}. \tag{19}$$

Here we assume $E_{kt}$ is standard Gaussian and $E_{1t}, E_{2t}, .., E_{nt}$ are mutually independent and independent from $\mathbf{z}_{t-1}$. $\frac{1}{b_k}$, which depends on $\mathbf{z}_{t-1}$, is the standard deviation of the noise in $z_{kt}$. (For conciseness, we drop the argument of $b_k$ and $q_k$ when there is no confusion.) Note that in this model, if $q_k$ is 0 for all $k = 1, 2, ..., n$, it reduces to a multiplicative noise model. The identifiability result of $\mathbf{z}_t$ is established in the following proposition.

**Corollary S1** (Identifiablity under Heterogeneous Noise). *Suppose $\mathbf{x}_t$ was generated according to Eq. 10 and Eq. 19. Suppose Eq. 11 holds true. If $b_k \cdot \frac{\partial b_k}{\partial \mathbf{z}_{t-1}}$ and $b_k \cdot \frac{\partial b_k}{\partial \mathbf{z}_{t-1}}(z_{kt} - q_k) - b_k^2 \cdot \frac{\partial q_k}{\partial \mathbf{z}_{t-1}}$, with $k = 1, 2, ..., n$, which are in total $2n$ function vectors in $\mathbf{z}_{t-1}$, are linearly independent, then $\mathbf{z}_t$ must be an invertible, component-wise transformation of a permuted version of $\hat{\mathbf{z}}_t$.*

*Proof.* Under the assumptions, one can see that

$$\eta_{kt} = \log p(z_{kt} \,|\, \mathbf{z}_{t-1}) = -\frac{1}{2}\log(2\pi) + \log b_k - \frac{b_k^2}{2}(z_{kt} - q_k)^2.$$

Consequently, one can find

$$\frac{\partial^3 \eta_{kt}}{\partial z_{kt}^2 \partial z_{l,t-1}} = -b_k \cdot \frac{\partial b_k}{\partial z_{l,t-1}},$$

$$\frac{\partial^2 \eta_{kt}}{\partial z_{kt} \partial z_{l,t-1}} = -b_k \cdot \frac{\partial b_k}{\partial z_{l,t-1}}(z_{kt} - q_k) + b_k^2 \cdot \frac{\partial q_k}{\partial z_{l,t-1}}.$$

Then the linear independence of $\mathbf{v}_{kt}$ and $\mathring{\mathbf{v}}_{kt}$ (defined in Eq. 12), with $k = 1., 2, ..., n$, reduces to the linear independence condition in this proposition. Theorem S1 then implies that $\mathbf{z}_t$ must be an invertible, component-wise transformation of a permuted version of $\hat{\mathbf{z}}_t$. $\qquad\square$

Let us then consider another special case, with linear, non-Gaussian temporal model for $\mathbf{z}_t$: the latent processes follow Eq. 18, with $\mathbf{q}$ being a linear transformation and $E_{kt}$ following a particular class of non-Gaussian distributions. The following corollary shows that $\mathbf{z}_t$ is identifiable as long as each $z_{kt}$ receives causal influences from some components of $\mathbf{z}_{t-1}$.

**Corollary S2** (Identifiablity under a Specific Linear, Non-Gaussian Model for Latent Processes). *Suppose $\mathbf{x}_t$ was generated according to Eq. 10 and Eq. 18, in which $\mathbf{q}$ is a linear transformation and for each $z_{kt}$, there exists at least one $k'$ such that $c_{kk'} \triangleq \frac{\partial z_{kt}}{\partial z_{k',t-1}} \neq 0$. Assume the noise term $E_{kt}$ follows a zero-mean generalized normal distribution:*

$$p(E_{kt}) \propto e^{-\lambda |e_{kt}|^\beta}, \quad \text{with positive } \lambda \text{ and } \beta > 2 \text{ and } \beta \neq 3. \tag{20}$$

*Suppose Eq. 11 holds true. Then $\mathbf{z}_t$ must be an invertible, component-wise transformation of a permuted version of $\hat{\mathbf{z}}_t$.*

*Proof.* In this case, we have

$$\frac{\partial^3 \eta_{kt}}{\partial z_{kt}^2 \partial z_{k',t-1}} = -\lambda \cdot \mathrm{sgn}(e_{kt}) \cdot \alpha(\beta - 1)(\beta - 2)|e_{kt}|^{\beta-3}c_{kk'}, \tag{21}$$

$$\frac{\partial^2 \eta_{kt}}{\partial z_{kt} \partial z_{k',t-1}} = -\lambda\beta(\beta - 1)|e_{kt}|^{\beta-2}c_{kk'}.$$

We know that $|e_{lt}|^{\beta-2}$ and $|e_{lt}|^{\beta-3}$ are linearly independent (because their ratio, $|e_{lt}|$, is not constant). Furthermore, $|e_{lt}|^{\beta-2}$ and $|e_{lt}|^{\beta-3}$, with $l = 1, 2, ..., n$, are $2n$ linearly independent functions (because of the different arguments involved).

Suppose there exist $\alpha_{l1}$ and $\alpha_{l2}$, with $l = 1, 2, ..., n$, such that

$$\sum_{l=1}^n \left( \alpha_{l1}\mathbf{v}_{lt} + \alpha_{l2}\mathring{\mathbf{v}}_{lt} \right) = 0. \tag{22}$$

It is assumed that for each $k = 1, 2, ..., n$, there exists at least one $k'$ such that $c_{kk'} \neq 0$. Eq. 22 then implies that for any $k$ we have

$$\alpha_{k1}c_{kk'}|e_{kt}|^{\beta-2} + \alpha_{k2}c_{kk'}|e_{kt}|^{\beta-3} + \sum_{l \neq k}\left( \alpha_{l1}c_{lk'}|e_{lt}|^{\beta-2} + \alpha_{l2}c_{lk'}|e_{lt}|^{\beta-3} \right) = 0. \tag{23}$$

Since $|e_{lt}|^{\beta-2}$ and $|e_{lt}|^{\beta-3}$, with $l = 1, 2, ..., n$, are linearly independent and $c_{kk'} \neq 0$, to make the above equation hold, one has to set $\alpha_{k1} = \alpha_{k2} = 0$. As this applies to any $k$, we know that for Eq. 22 to be satisfied, $\alpha_{l1}$ and $\alpha_{l2}$ must be 0, for all $l = 1, 2, ..., n$. That is, $\mathbf{v}_{1t}, \mathring{\mathbf{v}}_{1t}, \mathbf{v}_{2t}, \mathring{\mathbf{v}}_{2t}, ..., \mathbf{v}_{nt}, \mathring{\mathbf{v}}_{nt}$ are linearly independent. The linear independence condition in Theorem S1 is satisfied. Therefore $\mathbf{z}_t$ must be an invertible, component-wise transformation of a permuted version of $\hat{\mathbf{z}}_t$. $\qquad\square$

## S1.2 Proof for Theorem 2 and 3

Let $\mathbf{v}_{kt}(u_r)$ be $\mathbf{v}_{kt}$, which is defined in Eq. 12, in the $u_r$ context. Similarly, Let $\mathring{\mathbf{v}}_{kt}(u_r)$ be $\mathring{\mathbf{v}}_{kt}$ in the $u_r$ context. Let

$$\mathbf{s}_{kt} \triangleq \left( \mathbf{v}_{kt}(u_1)^\intercal, ..., \mathbf{v}_{kt}(u_m)^\intercal, \frac{\partial^2 \eta_{kt}(u_2)}{\partial z_{kt}^2} - \frac{\partial^2 \eta_{kt}(u_1)}{\partial z_{kt}^2}, ..., \frac{\partial^2 \eta_{kt}(u_m)}{\partial z_{kt}^2} - \frac{\partial^2 \eta_{kt}(u_{m-1})}{\partial z_{kt}^2} \right)^\intercal,$$

$$\mathring{\mathbf{s}}_{kt} \triangleq \left( \mathring{\mathbf{v}}_{kt}(u_1)^\intercal, ..., \mathring{\mathbf{v}}_{kt}(u_m)^\intercal, \frac{\partial \eta_{kt}(u_2)}{\partial z_{kt}} - \frac{\partial \eta_{kt}(u_1)}{\partial z_{kt}}, ..., \frac{\partial \eta_{kt}(u_m)}{\partial z_{kt}} - \frac{\partial \eta_{kt}(u_{m-1})}{\partial z_{kt}} \right)^\intercal.$$

As provided below, in our case, the identifiablity of $\mathbf{z}_t$ is guaranteed by the linear independence of the whole function vectors $\mathbf{s}_{kt}$ and $\mathring{\mathbf{s}}_{kt}$, with $k = 1, 2, ..., n$. However, the identifiability result in Yao et al. (2021) relies on the linear independence of only the last $m - 1$ components of $\mathbf{s}_{kt}$ and $\mathring{\mathbf{s}}_{kt}$ with $k = 1, 2, ..., n$; this linear independence is generally a much stronger condition.

**Theorem S2** (Identifiability under Changing Causal Dynamics). *Suppose the observed processes $\mathbf{x}_t$ was generated by Eq. 10 and that the conditional distribution $p(z_{kt} \,|\, \mathbf{z}_{t-1})$ may change across $m$ values of the context variable $\mathbf{u}$, denoted by $u_1, u_2, ..., u_m$. Suppose the components of $\mathbf{z}_t$ are mutually independent conditional on $\mathbf{z}_{-1}$ in each context. Assume that the components of $\hat{\mathbf{z}}_t$ produced by Eq. 11 are also mutually independent conditional on $\hat{\mathbf{z}}_{t-1}$. If the $2n$ function vectors $\mathbf{s}_{kt}$ and $\mathring{\mathbf{s}}_{kt}$, with $k = 1, 2, ..., n$, are linearly independent, then $\hat{\mathbf{z}}_t$ is a permuted invertible component-wise transformation of $\mathbf{z}_t$.*

*Proof.* As in the proof of Theorem S1, because the components of $\hat{\mathbf{z}}_t$ are mutually independent conditional on $\hat{\mathbf{z}}_{t-1}$, we know that for $i \neq j$,

$$\frac{\partial^2 \log p(\hat{\mathbf{z}}_t \,|\, \hat{\mathbf{z}}_{t-1}; \mathbf{u})}{\partial \hat{z}_{it} \partial \hat{z}_{jt}} = \sum_{k=1}^{n} \left( \frac{\partial^2 \eta_{kt}(\mathbf{u})}{\partial z_{kt}^2} \cdot \mathbf{H}_{kit} \mathbf{H}_{kjt} + \frac{\partial \eta_{kt}(\mathbf{u})}{\partial z_{kt}} \cdot \frac{\partial \mathbf{H}_{kit}}{\partial \hat{z}_{jt}} \right) - \frac{\partial^2 \log |\mathbf{H}_t|}{\partial \hat{z}_{it} \partial \hat{z}_{jt}} \equiv 0. \quad (24)$$

Compared to Eq. 16, here we allow $p(\hat{\mathbf{z}}_t \,|\, \hat{\mathbf{z}}_{t-1})$ to depend on $\mathbf{u}$. Since the above equation is always 0, taking its partial derivative w.r.t. $z_{l,t-1}$ gives

$$\frac{\partial^3 \log p(\hat{\mathbf{z}}_t \,|\, \hat{\mathbf{z}}_{t-1}; \mathbf{u})}{\partial \hat{z}_{it} \partial \hat{z}_{jt} \partial z_{l,t-1}} = \sum_{k=1}^{n} \left( \frac{\partial^3 \eta_{kt}(\mathbf{u})}{\partial z_{kt}^2 \partial z_{l,t-1}} \cdot \mathbf{H}_{kit} \mathbf{H}_{kjt} + \frac{\partial^2 \eta_{kt}(\mathbf{u})}{\partial z_{kt} \partial z_{l,t-1}} \cdot \frac{\partial \mathbf{H}_{kit}}{\partial \hat{z}_{jt}} \right) \equiv 0. \quad (25)$$

Similarly, Using different values for $\mathbf{u}$ in Eq. 24 take the difference of this equation across them gives

$$\frac{\partial^2 \log p(\hat{\mathbf{z}}_t \,|\, \hat{\mathbf{z}}_{t-1}; u_{r+1})}{\partial \hat{z}_{it} \partial \hat{z}_{jt}} - \frac{\partial^2 \log p(\hat{\mathbf{z}}_t \,|\, \hat{\mathbf{z}}_{t-1}; u_{r+1})}{\partial \hat{z}_{it} \partial \hat{z}_{jt}}$$

$$= \sum_{k=1}^{n} \left[ \left( \frac{\partial^2 \eta_{kt}(u_{r+1})}{\partial z_{kt}^2} - \frac{\partial^2 \eta_{kt}(u_r)}{\partial z_{kt}^2} \right) \cdot \mathbf{H}_{kit} \mathbf{H}_{kjt} + \left( \frac{\partial \eta_{kt}(u_{r+1})}{\partial z_{kt}} - \frac{\partial \eta_{kt}(u_r)}{\partial z_{kt}} \right) \cdot \frac{\partial \mathbf{H}_{kit}}{\partial \hat{z}_{jt}} \right] \equiv 0.$$

$$(26)$$

Therefore, if $\mathbf{s}_{kt}$ and $\mathring{\mathbf{s}}_{kt}$, for $k = 1, 2, ..., n$, are linearly independent, $\mathbf{H}_{kit} \mathbf{H}_{kjt}$ has to be zero for all $k$ and $i \neq j$. Then as shown in the proof of Theorem S1, $\hat{\mathbf{z}}_t$ must be a permuted component-wise invertible transformation of $\mathbf{z}_t$. □

**Theorem S3** (Identifiability under Observation Changes). *Suppose $\mathbf{x}_t = \mathbf{g}(\mathbf{z}_t)$ and that the conditional distribution $p(z_{k,t} \,|\, \mathbf{u})$ may change across $m$ values of the context variable $\mathbf{u}$, denoted by $u_1, u_2, ..., u_m$. Suppose the components of $\mathbf{z}_t$ are mutually independent conditional on $\mathbf{u}$ in each context. Assume that the components of $\hat{\mathbf{z}}_t$ produced by Eq. 3 are also mutually independent conditional on $\hat{\mathbf{z}}_{t-1}$. If the $2n$ function vectors $\mathbf{s}_{k,t}$ and $\hat{\mathbf{s}}_{k,t}$, with $k = 1, 2, ..., n$, are linearly independent, then $\hat{\mathbf{z}}_t$ is a permuted invertible component-wise transformation of $\mathbf{z}_t$.*

*Proof.* As in the proof of Theorem S2, because $\mathbf{z}_t$ is not dependent on the history $\mathbf{z}_{t-1}$ so are the components of $\hat{\mathbf{z}}_t$, the conditioning on $\hat{\mathbf{z}}_t$ in Eq. 24 and the following equations can be removed because of the independence. This directly leads to the same conclusion as in Theorem S2.

□

**Corollary S3** (Identifiability under Modular Distribution Shifts). *Assume the data generating process in Eq. 1. If the three partitioned latent components* $\mathbf{z}_t = (\mathbf{z}_t^{fix}, \mathbf{z}_t^{chg}, \mathbf{z}_t^{obs})$ *respectively satisfy the conditions in **Theorem 1**, **Theorem 2**, and **Theorem 3**, then* $\mathbf{z}_t$ *must be an invertible, component-wise transformation of a permuted version of* $\hat{\mathbf{z}}_t$.

*Proof.* Because the three partitioned subspaces $(\mathbf{z}_t^{\text{fix}}, \mathbf{z}_t^{\text{chg}}, \mathbf{z}_t^{\text{obs}})$ are conditional independent given the history and domain index, it is straightforward to factorize the joint conditional log density into three components. By using the proof in Theorem 1, 2, and 3, we can directly derive the same quantity as in Eq. 15 or Eq. 24. Therefore, if $\mathbf{s}_{kt}$ and $\mathring{\mathbf{s}}_{kt}$, for $k = 1, 2, ..., n$, are linearly independent, $\mathbf{H}_{kit}\mathbf{H}_{kjt}$ has to be zero for all $k$ and $i \neq j$. Then as shown in the proof of Theorem S1, $\hat{\mathbf{z}}_t$ must be a permuted component-wise invertible transformation of $\mathbf{z}_t$. $\square$

## S1.3 Comparisons with Existing Nonlinear ICA Theories

We compare our established theory with (1) PCL [7], (2) SlowVAE [12], (3) i-VAE [9], (4) GCL [8] and (5) LEAP [14] in terms of their mathematical formulation and assumptions.

**PCL [7]**    The formulation of the underlying processes in PCL is in Eq. 27:

$$\log p(z_{i,t}|z_{i,t-1}) = G(z_{i,t} - \rho z_{i,t-1}) \quad \text{or} \quad \log p(z_{i,t}|z_{i,t-1}) = -\lambda \left(z_{i,t} - r(z_{i,t-1})\right)^2 + \text{const}, \quad (27)$$

where $G$ is a non-quadratic function corresponding to the log-pdf of innovations, $\rho < 1$ is the regression coefficient, $r$ is some nonlinear, strictly monotonic regression, and $\lambda$ is a positive precision parameter.

PCL is applicable to **stationary environments** only when the sources $z_{it}$ are mutually independent (see Assumption 1 of Theorem 1 in PCL) and follow functional and distribution assumptions in Eq. 27. Our formulation allows latent variables to have arbitrary, nonparametric time-delayed causal relations in between without functional form or distribution assumptions.

**SlowVAE [12]**    The formulation of the underlying sources in SlowVAE is in Eq. 28:

$$p(\mathbf{z}_t|\mathbf{z}_{t-1}) = \prod_{i=1}^{d} \frac{\alpha\lambda}{2\Gamma(1/\alpha)} \exp-(\lambda|z_{i,t} - z_{i,t-1}|^\alpha) \quad with \quad \alpha < 2, \quad (28)$$

where the latent processes have independent, identity transitions with generalized Laplacian noises.

SlowVAE established identifiability under stationary, mutually independent processes (similar to PCL [7]), in which time-delayed causal influences are not allowed. Furthermore, it assumes that the transition function of each independent process is an identity function and the process noise has generalized Laplacian distribution. Our **Theorem 1** includes both SlowVAE and PCL as special cases, in the sense that (1) we remove the functional and distributional assumptions to allow the latent processes to have nonparametric causal influences in between, and (2) our **Corollay 2**, which is an illustrative example of **Theorem 1**, further completes to Eq. 28 by allowing **linear time-delayed transitions** in the latent process with **non-Gaussian noises**.

**i-VAE [9]**    Similar to TCL [6] and GIN [10], i-VAE exploits the nonstationarity brought by class labels on the distribution of latent variables. As one can see from Eq. 29, the latent variables are conditionally independent, without causal relations in between while all of our theorems consider (time-delayed) causal relations between latent variables. In addition, iVAE assumes the modulation of class labels on latent distributions is limited within the exponential family distribution. On the contrary, our nonparametric conditions (Theorems 1,2,3) allow any kind of modulation caused by fixed, changing transition dynamics or observation changes without those strong assumptions on the distribution of latent variables or noise distribution (i.e., SlowVAE [12]).

$$p_{T,\lambda}(\mathbf{z}|\mathbf{u}) = \prod_i \frac{Q_i(z_i)}{z_i(\mathbf{u})} \exp\left[\sum_{j=1}^{k} T_{i,j}(z_i)\lambda_{i,j}(\mathbf{u})\right] \quad (29)$$

**GCL [8]** The formulation of the underlying sources in GCL is in Eq 30, which is also described by Eqs. 4,15 in the original paper [8]:

$$p(\mathbf{z}_t|\mathbf{z}_{t-1}) = \prod_{i=1}^{d} p_i(z_{i,t}|z_{i,t-1}), \tag{30}$$

where the latent processes are free from the exponential family distribution assumptions but still constrained within mutually-independent processes. On the contrary, our work considers **causally-related** latent space in which cross causal relations between latent variables can be recovered. Additionally, we want to mention that we provide an antenna tube in the schema of the proof in **Theorems** 1-2-3, which is a more direct way of using sufficient variability conditions than [8].

**LEAP [14]** LEAP in Eq. 31 considers one special case of nonstationarity caused by changes in noise distributions, while our work can allow changing causal relations over context. Furthermore, because LEAP assumes all latent processes are changed across contexts, it doesn't use or benefit from the fixed time-delayed causal relations for identifiability. On the contrary, our work exploits the modular distribution changes from the fixed causal dynamics, changing dynamics, and observation changes, and hence our identifiability conditions are generally weaker than [14].

$$\underbrace{\mathbf{x}_t = g(\mathbf{z}_t)}_{\text{Nonlinear mixing}}, \quad \underbrace{z_{it} = f_i\left(\{z_{j,t-\tau}|z_{j,t-\tau} \in \mathbf{Pa}(z_{it})\}, \epsilon_{it}\right)}_{\text{Nonparametric transition}} \; with \; \underbrace{\epsilon_{it} \sim p_{\epsilon_i|\mathbf{u}}}_{\text{Nonstationary noise}} \, . \tag{31}$$

## S1.4 Discussion of the Assumptions

We first explain and justify each critical assumption in the proposed conditions. We then discuss how restrictive or mild the conditions are in real applications.

### S1.4.1 Linear Independence Condition

Our proposed linear independence condition is a combination of stationary identifiability conditions (Eq. 4) in each context $u_r$, plus the identifiability conditions for nonrecurrent influences (Eq. 8). The condition is essential to make each row of the Jacobian matrix $\mathbf{H}_t$ of the indeterminacy function of the learned latent space in Eq. 17 to have only one non-zero entry, thus making the learned latent variables identifiable up to permutation and component-wise invertible transformations.

In **stationary environments**, this condition essentially states that, during the generation of $\mathbf{z}_t$, if either of the two conditions is satisfied, then the linear independence condition holds in general.

**(1)** If the history information $\mathbf{z}_{\text{Hx}} = \{\mathbf{z}_{t-\tau}\}_{\tau=1}^{L}$ up to maximum time lag $L$, and the process noise $\epsilon_t$ are coupled in a nontrivial way for generating $\mathbf{z}_t$ (e.g., heterogeneous noise process in Eq. 32), such that $\mathbf{z}_{\text{Hx}}$ can modulate the variance or higher-order statistics of the conditional distribution $p(\mathbf{z}_t|\mathbf{z}_{\text{Hx}})$, then the linear independence condition generally holds;

$$z_{k,t} = q_k(\mathbf{z}_{t-1}) + \frac{1}{b_k(\mathbf{z}_{t-1})}\epsilon_{k,t}. \tag{32}$$

**(2)** If the latent transition is an additive noise model (then (1) is violated) but the process noise is non-Gaussian, it will be extremely hard for the linear independence condition to be violated. Roughly speaking, for a randomly chosen conditional density function $p(z_{k,t}|\mathbf{z}_{t-1})$ in which $z_{k,t}$ is not independent from $\mathbf{z}_{t-1}$ (i.e., there is temporal dependence in the latent processes) and which does not follow an additive noise model with Gaussian noise, the chance for its specific second- and third-order partial derivatives to be linearly dependent is slim.

In **nonstationary environments**, this condition was introduced in GCL [8], namely, "sufficient variability", to extend the modulated exponential families [6] to general modulated distributions. Essentially, the condition says that the nonstationary regimes $\mathbf{u}$ must have a sufficiently complex and diverse effect on the transition distributions. In other words, if the underlying distributions are composed of relatively many domains of data, the condition generally holds true. For instance, in the linear Auto-Regressive (AR) model with Gaussian innovations where only the noise variance

changes, the condition reduces to the statement in [42] that the variance of each noise term fluctuates somewhat independently of each other in different nonstationary regimes. Then the condition is easily attained if the variance vector of noise terms in any regime is not a linear combination of variance vectors of noise terms in other regimes.

We further illustrate the condition using the example of modulated conditional exponential families in [8]. Let the log-pdf $q(\mathbf{z}_t|\{\mathbf{z}_{t-\tau}\}, \mathbf{u})$ be a conditional exponential family distribution of order $k$ given nonstationary regime $\mathbf{u}$ and history $\mathbf{z}_{\text{Hx}} = \{\mathbf{z}_{t-\tau}\}$:

$$q(z_{it}|\mathbf{z}_{\text{Hx}}, \mathbf{u}) = q_i(z_{it}) + \sum_{j=1}^{k} q_{ij}(z_{it})\lambda_{ij}(\mathbf{z}_{\text{Hx}}, \mathbf{u}) - \log Z(\mathbf{z}_{\text{Hx}}, \mathbf{u}), \qquad (33)$$

where $q_i$ is the base measure, $q_{ij}$ is the function of the sufficient statistic, $\lambda_{ij}$ is the natural parameter, and $\log Z$ is the log-partition. Loosely speaking, the sufficient variability holds if the modulation of by $\mathbf{u}$ on the conditional distribution $q(z_{it}|\mathbf{z}_{\text{Hx}}, \mathbf{u})$ is not too simple in the following sense:

1. Higher order of $k$ ($k > 1$) is required. If $k = 1$, the sufficient variability cannot hold;

2. The modulation impacts $\lambda_{ij}$ by $\mathbf{u}$ must be linearly independent across regimes $\mathbf{u}$. The sufficient statistics functions $q_{ij}$ cannot be all linear, i.e., we require higher-order statistics.

Further details of this example can be found in Appendix B of [8]. In summary, we need the modulation by $\mathbf{u}$ to have diverse (i.e., distinct influences) and complex impacts on the underlying data generation process.

**Applicability** By combining the stationary and nonstationary conditions, our proposed identifiability condition is generally mild, in the sense that if there is at least one regime $r$ out of the $m$ contexts which satisfies the stationary identifiability conditions, OR, if the overall nonstationary influences are diverse and complex, thus satisfying the nonstationary identifiability conditions, the latent temporal causal processes are identifiable. For stationary conditions, the only situation where we find the latent processes unidentifiable is when the latent temporal transition is described by a Gaussian additive noise model, which violates both (1) and (2). However, for real-world data, it is very unlikely for the process noise to be perfectly Gaussian. Nonstationarity seems to be prominent in many kinds of temporal data. For example, nonstationary variances are seen in EEG/MEG, and natural video, and are closely related to changes in volatility in financial time series [6]. The data that most likely satisfy the nonstationary condition is a collection of multiple trials/segments of data with different temporal dynamics in between.

### S1.4.2 Independent Noise Condition

The IN condition was introduced in the Structural Equation Model (SEM), which represents effect $Y$ as a function of direct causes $X$ and noise $E$:

$$Y = f(X, E) \quad with \quad \underbrace{X \perp\!\!\!\perp E}_{\text{IN condition}}. \qquad (34)$$

If $X$ and $Y$ do not have a common cause, as seen from the causal sufficiency assumption of structural equation models in Chapter 1.4.1 of Pearl's book [43], the IN condition states that the unexplained noise variable $E$ is statistically independent of cause $X$. IN is a direct result of assuming causal sufficiency in SEM. The main idea for the proof is that if IN is violated, then by the common cause principle [44], there exist hidden confounders that cause their dependence, thus violating the causal sufficiency assumption. Furthermore, the noise terms in different variables are mutually independent for a causally sufficient system with acyclic causal relations. The main idea is that when the noise terms are dependent, it is customary to encode such dependencies by augmenting the graph with hidden confounder variables [43], which means that the system is not causally sufficient.

This paper assumes that the underlying latent processes form a casually-sufficient system without latent causal confounders. Then, the process noise terms $\epsilon_{it}$ are mutually independent, and moreover, the process noise terms $\epsilon_{it}$ are independent of direct cause/parent nodes $\mathbf{Pa}(z_{it})$ because of time information (the causal graph is acyclic because of the temporal precedence constraint).

**Applicability**   Loosely speaking, if there are no latent causal confounders in the (latent) causal processes and the sampling frequency is high enough to observe the underlying dynamics, then the IN condition assumed in this paper is satisfied in a causally-sufficient system and, moreover, there is no instantaneous causal influence (because of the high enough resolution). At the same time, we acknowledge that there exist situations where the resolution is low and there appears to be instantaneous dependence. However, several pieces of work deal with causal discovery from measured time series in such situations; see. e.g., [45, 46, 47, 48]. In case there are instantaneous causal relations among latent causal processes, one would need additional sparsity or minimality conditions to recover the latent processes and their relations, as demonstrated in [49, 50]. How to address the issue of instantaneous dependency or instantaneous causal relations in the latent processes will be one line of our future work.

### S1.4.3   Causal Influences between Observed Variables

Although causal discovery between observed variables is not the main focus of our work, our model can discover causal relations between observed variables as a special case, thanks to the nonlinear mixing function assumed in this paper. In our formulation in Eq. 1, we assume the observations $x_t$ are nonlinear, invertible mixtures of latent processes $z_t$. However, if in the data generating process, one observed variable $x_i$ has direct causal edges with the other observed variable $x_j$, the mixing function that generates $x_i$ and $x_j$ will just be reduced to identity mappings of $z_i$ and $z_j$, which is a special case of the nonlinear invertible mixing function in Eq. 1.

### S1.5   Extension to Multiple Time Lags

For the sake of simplicity, we consider one lag for the latent processes in Section 3 (and only in Section 3). Our identifiability proof can actually be applied for arbitrary lags directly. For instance, in the stationary case in Eq. 4, one can simply re-define $\eta_{kt} \triangleq \log p(z_{k,t}|\mathbf{z}_{Hx})$, where $\mathbf{z}_{Hx}$ denotes the lagged latent variables up to maximum time lag $L$. We plug it into Eq. 4, and take derivatives with regard to $z_{1,t-\tau}, \ldots, z_{n,t-\tau}$, which can be any latent temporal variables at lag $\tau$, instead of $z_{1,t-1}, \ldots, z_{n,t-1}$. If there exists one $\tau$ (out of the $L$ lags) that satisfies the condition, then the stationary latent processes are identifiable. Similarly, for Eq. 8, one can simply re-define $\eta\_kt(u_r) \triangleq \log p(z_{k,t}|\mathbf{z}_{Hx}, u_r)$ and plug it into Eq. 8. No extra changes are needed.

## S2   Experiment Settings

### S2.1   Datasets

### S2.1.1   Synthetic Dataset Generation

We consider three representative simulation settings to validate the identifiability results under fixed causal dynamics, changing causal dynamics, and modular distribution shift which contains fixed dynamics, changing dynamics, and global changes together in the latent processes. For synthetic datasets with fixed and changing causal dynamics, we set latent size $n = 8$. For the modular shift dataset, we add one dimension for global observation changes. The lag number of the process is set to $L = 2$. The mixing function $g$ is a random three-layer MLP with LeakyReLU units.

**Fixed Causal Dynamics**   For the fixed causal dynamics. We generate 100,000 data points according to Eq. (6), where the latent size is $n = 8$, lag number of the process is $L = 2$. We apply a 2-layer MLP with LeakyReLU as the state transition function. The process noise are sampled from i.i.d. Gaussian distribution ($\sigma = 0.1$). The process noise terms are coupled with the history information through multiplication with the average value of all the time-lagged latent variables.

**Changing Causal Dynamics**   We use a Gaussian additive noise model with changes in the influencing strength as the latent processes. To add changes, we vary the values of the first layer of the MLP across the 20 segments and generate 7,500 samples for each segment. The entries of the kernel matrix of the first layer are uniformly distributed between $[-1, 1]$ in each domain.

**Modular Distribution Shifts**   The latent space of this dataset is partitioned into 6 fixed dynamics components under the heterogeneous noise model, 2 changing components with changing causal

dynamics and 1 component modulated by domain index only. The fixed and changing dynamics components follow the same generating procedures above. The global change component is sampled from i.i.d Gaussian distribution whose mean and variance are modulated by domain index. In particular, distribution mean terms are uniformly sampled between $[-1, 1]$ and variance terms are uniformly sampled between $[0.01, 1]$.

### S2.1.2    Real-world Dataset

**Modified Cartpole**    The Cartpole problem [30] "consists of a cart and a vertical pendulum attached to the cart using a passive pivot joint. The cart can move left or right. The task is to prevent the vertical pendulum from falling by putting a force on the cart to move it left or right. The action space consists of two actions: moving left or right."

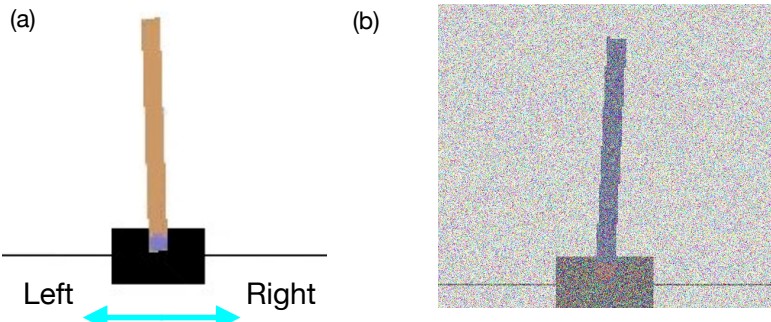

Figure S1: "Visual examples of Cartpole game and change factors. (a) Cartpole game; (b) Modified Cartpole game with Gaussian noise on the image. The light blue arrows are added to show the direction in which the agent can move." Figure source: [30].

The original dataset [30] introduces "two change factors respectively for the state transition dynamics $\theta_k^{\text{dyn}}$: varying gravity and varying mass of the cart, and a change factor in the observation function $\theta_k^{\text{obs}}$ that is the image noise level. Fig. S1 gives a visual example of Cartpole game, and the image with Gaussian noise. The images of the varying gravity and mass look exactly like the original image. Specifically, in the gravity case, we consider source domains with gravity $g = \{5, 10, 20, 30, 40\}$. We take into account both interpolation (where the gravity in the target domain is in the support of that in source domains) with $g = \{15\}$, and extrapolation (where it is out of the support w.r.t. the source domains) with $g = \{55\}$. Similarly, we consider source domains where the mass of the cart is $m = \{0.5, 1.5, 2.5, 3.5, 4.5\}$, while in target domains it is $m = \{1.0, 5.5\}$. In terms of changes on the observation function, we add Gaussian noise on the images with variance $\sigma = \{0.25, 0.75, 1.25, 1.75, 2.25\}$ in source domains, and $\sigma = \{0.5, 2.75\}$ in target domains. The detailed settings in both source and target domains are in Table S1."

|  | Gravity | Mass | Noise |
|---|---|---|---|
| Source domains | $\{5, 10, 20, 30, 40\}$ | $\{0.5, 1.5, 2.5, 3.5, 4.5\}$ | $\{0.25, 0.75, 1.25, 1.75, 2.25\}$ |
| Interpolation set | $\{15\}$ | $\{1.0\}$ | $\{0.5\}$ |
| Extrapolation set | $\{55\}$ | $\{5.5\}$ | $\{2.75\}$ |

Table S1: "The settings of source and target domains for modified Cartpole experiments" [30].

**CMU-Mocap**    CMU MoCap (http://mocap.cs.cmu.edu/) is an open-source human motion capture dataset with various motion capture recordings (e.g., walk, jump, basketball, etc.) performed by over 140 subjects. In this work, we fit our model on 11 trials of "walk" recordings (Subject #8). Skeleton-based measurements have 62 observed variables corresponding to the locations of joints (e.g., head, foot, shoulder, wrist, throat, etc.) of the human body at each time step.

## S2.2 Mean Correlation Coefficient

MCC is a standard metric for evaluating the recovery of latent factors in ICA literature. MCC first calculates the absolute values of the correlation coefficient between every ground-truth factor against every estimated latent variable. Pearson correlation coefficients or Spearman's rank correlation coefficients can be used depending on whether componentwise invertible nonlinearities exist in the recovered factors. The possible permutation is adjusted by solving a linear sum assignment problem in polynomial time on the computed correlation matrix.

# S3 Implementation Details

## S3.1 Modular Prior Likelihood Derivation

Let us start with an illustrative example of stationary latent causal processes consisting of two time-delayed latent variables, i.e., $\mathbf{z}_t = [z_{1,t}, z_{2,t}]$ with maximum time lag $L = 1$, i.e., $z_{i,t} = f_i(\mathbf{z}_{t-1}, \epsilon_{i,t})$ with mutually independent noises. Let us write this latent process as a transformation map $\mathbf{f}$ (note that we overload the notation $f$ for transition functions and for the transformation map):

$$\begin{bmatrix} z_{1,t-1} \\ z_{2,t-1} \\ z_{1,t} \\ z_{2,t} \end{bmatrix} = \mathbf{f}\left(\begin{bmatrix} z_{1,t-1} \\ z_{2,t-1} \\ \epsilon_{1,t} \\ \epsilon_{2,t} \end{bmatrix}\right). \tag{35}$$

By applying the change of variables formula to the map $\mathbf{f}$, we can evaluate the joint distribution of the latent variables $p(z_{1,t-1}, z_{2,t-1}, z_{1,t}, z_{2,t})$ as:

$$p(z_{1,t-1}, z_{2,t-1}, z_{1,t}, z_{2,t}) = p(z_{1,t-1}, z_{2,t-1}, \epsilon_{1,t}, \epsilon_{2,t})/\left|\det \mathbf{J_f}\right|, \tag{36}$$

where $\mathbf{J_f}$ is the Jacobian matrix of the map $\mathbf{f}$, which is naturally a low-triangular matrix:

$$\mathbf{J_f} = \begin{bmatrix} 1 & 0 & 0 & 0 \\ 0 & 1 & 0 & 0 \\ \frac{\partial z_{1,t}}{\partial z_{1,t-1}} & \frac{\partial z_{1,t}}{\partial z_{2,t-1}} & \frac{\partial z_{1,t}}{\partial \epsilon_{1,t}} & 0 \\ \frac{\partial z_{2,t}}{\partial z_{1,t-1}} & \frac{\partial z_{2,t}}{\partial z_{2,t-1}} & 0 & \frac{\partial z_{2,t}}{\partial \epsilon_{2,t}} \end{bmatrix}.$$

Given that this Jacobian is triangular, we can efficiently compute its determinant as $\prod_i \frac{\partial z_{i,t}}{\partial \epsilon_{i,t}}$. Furthermore, because the noise terms are mutually independent, and hence $\epsilon_{i,t} \perp \epsilon_{j,t}$ for $j \neq i$ and $\epsilon_t \perp \mathbf{z}_{t-1}$, we can write the RHS of Eq. 36 as:

$$p(z_{1,t-1}, z_{2,t-1}, z_{1,t}, z_{2,t}) = p(z_{1,t-1}, z_{2,t-1}) \times p(\epsilon_{1,t}, \epsilon_{2,t})/\left|\det \mathbf{J_f}\right| \quad (\text{because } \epsilon_t \perp \mathbf{z}_{t-1})$$

$$= p(z_{1,t-1}, z_{2,t-1}) \times \prod_i p(\epsilon_{i,t})/\left|\det \mathbf{J_f}\right| \quad (\text{because } \epsilon_{1,t} \perp \epsilon_{2,t}) \tag{37}$$

Finally, by canceling out the marginals of the lagged latent variables $p(z_{1,t-1}, z_{2,t-1})$ on both sides, we can evaluate the transition prior likelihood as:

$$p(z_{1,t}, z_{2,t}|z_{1,t-1}, z_{2,t-1}) = \prod_i p(\epsilon_{i,t})/\left|\det \mathbf{J_f}\right| = \prod_i p(\epsilon_{i,t}) \times \left|\det \mathbf{J_f}^{-1}\right|. \tag{38}$$

Now we generalize this example and derive the modular prior likelihood below.

**Fixed Causal Dynamics** Let $\{f_s^{-1}\}_{s=1,2,3\dots}$ be a set of learned inverse fixed dynamics transition functions that take the estimated latent causal variables in the fixed dynamics subspace and lagged latent variables, and output the noise terms, i.e., $\hat{\epsilon}_{s,t} = f_s^{-1}\left(\hat{z}_{s,t}^{\text{fix}}, \{\hat{\mathbf{z}}_{t-\tau}\}\right)$.

Design transformation $\mathbf{A} \rightarrow \mathbf{B}$ with low-triangular Jacobian as follows:

$$\underbrace{\left[\hat{\mathbf{z}}_{t-L}, \ldots, \hat{\mathbf{z}}_{t-1}, \hat{\mathbf{z}}_t^{\text{fix}}\right]^\top}_{\mathbf{A}} \text{ mapped to } \underbrace{\left[\hat{\mathbf{z}}_{t-L}, \ldots, \hat{\mathbf{z}}_{t-1}, \hat{\epsilon}_{s,t}\right]^\top}_{\mathbf{B}}, \text{ with } \mathbf{J}_{\mathbf{A} \to \mathbf{B}} = \begin{pmatrix} \mathbb{I}_{nL} & 0 \\ * & \text{diag}\left(\frac{\partial f_{s,i}^{-1}}{\partial \hat{z}_{it}^{\text{fix}}}\right) \end{pmatrix}. \tag{39}$$

Similar to Eq. 38, we can obtain the joint distribution of the estimated fixed dynamics subspace as:

$$\log p(\mathbf{A}) = \underbrace{\log p\left(\hat{\mathbf{z}}_{t-L}, \ldots, \hat{\mathbf{z}}_{t-1}\right) + \sum_{i=1}^n \log p(\hat{\epsilon}_{s,t})}_{\text{Because of mutually independent noise assumption}} + \log\left(|\det\left(\mathbf{J}_{\mathbf{A} \to \mathbf{B}}\right)|\right). \tag{40}$$

$$\log p\left(\hat{\mathbf{z}}_t^{\text{fix}} | \{\hat{\mathbf{z}}_{t-\tau}\}_{\tau=1}^L\right) = \sum_{i=1}^n \log p(\hat{\epsilon}_{s,t}) + \sum_{i=1}^n \log\left|\frac{\partial f_s^{-1}}{\partial \hat{z}_{s,t}}\right| \tag{41}$$

**Changing Causal Dynamics**  The differences from fixed dynamics are that the learned inverse changing dynamics transition functions take additional learned change factors of the context as input arguments to out the noise terms, i.e., $\hat{\epsilon}_{c,t} = f_c^{-1}\left(\hat{z}_{c,t}^{\text{chg}}, \{\hat{\mathbf{z}}_{t-\tau}\}, \mathbf{u}_k\right) = f_c^{-1}\left(\hat{z}_{c,t}^{\text{chg}}, \{\hat{\mathbf{z}}_{t-\tau}\}, \boldsymbol{\theta}_k^{\text{dyn}}\right)$.

$$\log p\left(\hat{\mathbf{z}}_t^{\text{chg}} | \{\hat{\mathbf{z}}_{t-\tau}\}_{\tau=1}^L, \mathbf{u}_k\right) = \sum_{i=1}^n \log p(\hat{\epsilon}_{c,t} | \mathbf{u}_k) + \sum_{i=1}^n \log\left|\frac{\partial f_c^{-1}}{\partial \hat{z}_{c,t}}\right| \tag{42}$$

**Observation Changes**  The global observation changes are captured by the learned inverse $f_o^{-1}$, which takes the estimated latent subspace and the learned change factors for global observation $\boldsymbol{\theta}_k^{\text{obs}}$ of context $k$, and output random noise, i.e., $\hat{\epsilon}_{o,t} = f_o^{-1}\left(\hat{z}_{o,t}^{\text{obs}}, \mathbf{u}_k\right) = f_c^{-1}\left(\hat{z}_{c,t}^{\text{chg}}, \boldsymbol{\theta}_k^{\text{obs}}\right)$.

$$\log p\left(\hat{\mathbf{z}}_t^{\text{obs}} | \mathbf{u}_k\right) = \sum_{i=1}^n \log p(\hat{\epsilon}_{o,t} | \mathbf{u}_k) + \sum_{i=1}^n \log\left|\frac{\partial f_o^{-1}}{\partial \hat{z}_{o,t}}\right|. \tag{43}$$

### S3.2  Comparisons with AdaRL [30]

In terms of the implementation, in our work, we enforce the independent noise or conditional independence condition explicitly in Eq. 9 (derived in Appendix S3.1) for the identifiability of the latent processes. But disentanglement is not the main goal of AdaRL so it used Mixture Density Network (MDN) to approximate the transition prior. Our framework is simpler than AdaRL in the inference module (we use only $\mathbf{x}_t$ to infer $\mathbf{z}_t$) and the loss function (we don't have the prediction branch or the sparsity loss), thanks to the nonlinear ICA formulation in Eq. 1. Our identifiability conditions do not rely on the sparsity constraints in the underlying data generating process.

### S3.3  Network Architecture

We summarize our network architecture below and describe it in detail in Table S2 and Table S3.

### S3.4  Hyperparameter and Training

**Hyperparameter Selection**  The hyperparameters include $\beta$, which is the weight of KLD terms, as well as the latent size $n$ and maximum time lag $L$. We use the ELBO loss to select the best pair of $\beta$ because low ELBO loss always leads to high MCC. We always set a larger latent size than the true latent size. This is critical in real-world datasets because restricting the latent size will hurt the reconstruction performances and over-parameterization makes the framework robust to assumption violations. For the maximum time lag $L$, we set it by the rule of thumb. For instance, we use $L = 2$ for temporal datasets with a latent physics process (e.g, cartpole, cmu-mocap).

**Training Details**  The models were implemented in `PyTorch` 1.8.1. The VAE network is trained using AdamW optimizer for a maximum of 50 epochs and early stops if the validation ELBO loss does not decrease for five epochs. A learning rate of 0.002 and a mini-batch size of 64 are used. We have used three random seeds in each experiment and reported the mean performance with standard deviation averaged across random seeds.

Table S2: Architecture details. BS: batch size, T: length of time series, i_dim: input dimension, z_dim: latent dimension, LeakyReLU: Leaky Rectified Linear Unit.

| Configuration | Description | Output |
|---|---|---|
| **1. MLP-Encoder** | Encoder for Synthetic Data | |
| Input: $\mathbf{x}_{1:T}$ | Observed time series | BS $\times$ T $\times$ i_dim |
| Dense | 128 neurons, LeakyReLU | BS $\times$ T $\times$ 128 |
| Dense | 128 neurons, LeakyReLU | BS $\times$ T $\times$ 128 |
| Dense | 128 neurons, LeakyReLU | BS $\times$ T $\times$ 128 |
| Dense | Temporal embeddings | BS $\times$ T $\times$ z_dim |
| **2. MLP-Decoder** | Decoder for Synthetic Data | |
| Input: $\hat{\mathbf{z}}_{1:T}$ | Sampled latent variables | BS $\times$ T $\times$ z_dim |
| Dense | 128 neurons, LeakyReLU | BS $\times$ T $\times$ 128 |
| Dense | 128 neurons, LeakyReLU | BS $\times$ T $\times$ 128 |
| Dense | i_dim neurons, reconstructed $\hat{\mathbf{x}}_{1:T}$ | BS $\times$ T $\times$ i_dim |
| **5. Factorized Inference Network** | Bidirectional Inference Network | |
| Input | Sequential embeddings | BS $\times$ T $\times$ z_dim |
| Bottleneck | Compute mean and variance of posterior | $\mu_{1:T}, \sigma_{1:T}$ |
| Reparameterization | Sequential sampling | $\hat{\mathbf{z}}_{1:T}$ |
| **6. Modular Prior** | Nonlinear Transition Prior Network | |
| Input | Sampled latent variable sequence $\hat{\mathbf{z}}_{1:T}$ | BS $\times$ T $\times$ z_dim |
| InverseTransition | Compute estimated residuals $\hat{\epsilon}_{it}$ | BS $\times$ T $\times$ z_dim |
| JacobianCompute | Compute $\log\left(\left|\det\left(\mathbf{J}\right)\right|\right)$ | BS |

**Computing Hardware**  We used a machine with the following CPU specifications: Intel(R) Core(TM) i7-7700K CPU @ 4.20GHz; 8 CPUs, four physical cores per CPU, a total of 32 logical CPU units. The machine has two GeForce GTX 1080 Ti GPUs with 11GB GPU memory.

**Reproducibility**  We've included the code for the framework and all experiments in the supplement. We plan to release our code under the MIT License after the paper review period.

## S4  Additional Experiment Results

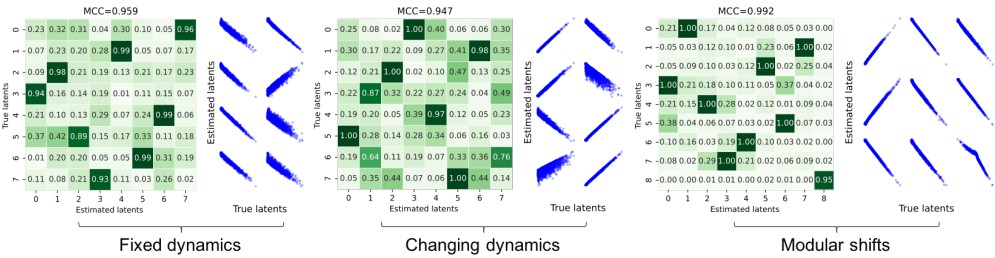

Figure S2: Results for three synthetic datasets: in each block, the left shows the MCC for causally-related and the left are scatterplots between estimated and true factors.

Table S3: Architecture details on CNN encoder and decoder. BS: batch size, T: length of time series, h_dim: hidden dimension, z_dim: latent dimension, F: number of filters, (Leaky)ReLU: (Leaky) Rectified Linear Unit.

| Configuration | Description | Output |
|---|---|---|
| **3.1.1 CNN-Encoder** | Feature Extractor | |
| Input: $\mathbf{x}_{1:T}$ | RGB video frames | BS $\times$ T $\times$ 3 $\times$ 64 $\times$ 64 |
| Conv2D | F: 32, BatchNorm2D, LeakyReLU | BS $\times$ T $\times$ 32 $\times$ 64 $\times$ 64 |
| Conv2D | F: 32, BatchNorm2D, LeakyReLU | BS $\times$ T $\times$ 32 $\times$ 32 $\times$ 32 |
| Conv2D | F: 32, BatchNorm2D, LeakyReLU | BS $\times$ T $\times$ 32 $\times$ 16 $\times$ 16 |
| Conv2D | F: 64, BatchNorm2D, LeakyReLU | BS $\times$ T $\times$ 64 $\times$ 8 $\times$ 8 |
| Conv2D | F: 64, BatchNorm2D, LeakyReLU | BS $\times$ T $\times$ 64 $\times$ 4 $\times$ 4 |
| Conv2D | F: 128, BatchNorm2D, LeakyReLU | BS $\times$ T $\times$ 128 $\times$ 1 $\times$ 1 |
| Dense | F: 2 * z_dim = dimension of hidden embedding | BS $\times$ T $\times$ 2 * z_dim |
| **4.1 CNN-Decoder** | Video Reconstruction | |
| Input: $\mathbf{z}_{1:T}$ | Sampled latent variable sequence | BS $\times$ T $\times$ z_dim |
| Dense | F: 128 , LeakyReLU | BS $\times$ T $\times$ 128 $\times$ 1 $\times$ 1 |
| ConvTranspose2D | F: 64, BatchNorm2D, LeakyReLU | BS $\times$ T $\times$ 64 $\times$ 4 $\times$ 4 |
| ConvTranspose2D | F: 64, BatchNorm2D, LeakyReLU | BS $\times$ T $\times$ 64 $\times$ 8 $\times$ 8 |
| ConvTranspose2D | F: 32, BatchNorm2D, LeakyReLU | BS $\times$ T $\times$ 32 $\times$ 16 $\times$ 16 |
| ConvTranspose2D | F: 32, BatchNorm2D, LeakyReLU | BS $\times$ T $\times$ 32 $\times$ 32 $\times$ 32 |
| ConvTranspose2D | F: 32, BatchNorm2D, LeakyReLU | BS $\times$ T $\times$ 32 $\times$ 64 $\times$ 64 |
| ConvTranspose2D | F: 3, estimated scene $\hat{\mathbf{x}}_{1:T}$ | BS $\times$ T $\times$ 3 $\times$ 64 $\times$ 64 |