# OpenReview forum: "Temporally Disentangled Representation Learning"
_NeurIPS.cc/2022/Conference — NeurIPS 2022 Accept_

### Official Review · Reviewer_7swA · 2022-06-28

**Rating:** 6
**Confidence:** 3
**Soundness:** 3 good
**Presentation:** 3 good
**Contribution:** 3 good

**Summary:**

The paper proposes a new framework for nonlinear Independent Component Analysis, LiLY, which recovers time-delayed latent causal variables from temporal data under stationary environments and under different distribution shifts. The authors provide proofs for the identifiability of the causal variables in the settings under consideration and show experimentally that their approach achieves improved performance on data that matches their assumptions.

**Questions:**

A number of questions regarding this paper are embedded in the "Strengths and Weaknesses" section above. Besides these questions, I would be interested in a more concrete connection between the proposed approach and causal discovery. If the authors can answer these points convincingly, as well as outline how and where they would embed their answers into the paper, I will consider raising my score.

**Limitations:**

I could not find any discussion on the limitations of the proposed method, and would appreciate if the authors could highlight some limitations in their rebuttal.

**Given the author's reply and revised version of the paper, I have increased my rating for this paper.**

**Strengths And Weaknesses:**

# Strengths
- The proposed framework consideres a novel setting for nonlinear ICA and provides convincing identiability results in terms of theory and practice.

- Overall, the paper follows a coherent structure and uses clear formulations for all theoretical and experimental settings.

- Table 1 gives a great overview that summarizes the novelty of the proposed framework compared to existing research.

# Weaknesses
1) In my eyes, the biggest weakness of this paper is that it does not provide any convincing motivation for the proposed approach. In the current form, the paper argues that existing approaches have a certain set of assumptions and that the proposed approach uses different assumptions. However, this is not really an argument in itself. What I would like to know is: why do these assumptions matter; how are existing approaches limited by the assumptions that they use and what new applications do you enable with your new approach? To embed these answers into the paper, I would propose a rewrite of the introduction. Additionally, it would be nice to include a discussion on the assumptions in section 3.1, i.e. why they are needed and how restrictive they are in practice. Besides this, the paper is also missing a discussion on the proposed separation of distribution changes into three types. Why did you choose these three types; are there any alternative separations that you considered and why is the proposed solution the best; are there any distribution changes that these types cannot (jointly) represent? Finally, a discussion on the limitations of the proposed approach is missing. Overall, this makes it very difficult to judge the significance of this work.

2) The connection between this method and causal discovery seems somewhat hand-wavy. The abstract claims that the proposed framework recovers latent variables and identifies their relations. While the experiments and theoretical results support the former claim, the latter seems to be missing from the paper, effectively removing the connection to causal discovery. Additionally, "causal dynamics" are mentioned throughout the paper without ever being defined.

3) According to l.297, all evaluations were conducted on the validation datasets. This is not appropriate according to standard machine learning practice. For convincing results, these experiments need to be repeated on the test datasets.

4) The paper's clarity could be improved in certain regards. For example, I found it confusing which variables and functions are estimated by the model and which ones are ground-truth factors. Does the model use the ground-truth transition functions, or does it estimate them? Does the model work with unknown distribution shifts as claimed in the abstract and intro, or does it require a domain index (l. 281) as input?



## Others
- Currently, the introduction also functions as a related works section. I would prefer a separation of these two sections. For one, this would allow the introduction to focus on the motivation for the proposed approach, as mentioned above. Additionally, a separate related work section could discuss existing work in more detail and provide an embedding of the proposed work into the existing research landscape - an aspect that is largely missing in the current form.

- There are grammatical errors and spelling mistakes throughout the paper that a standard spell checker should be able to catch.

- Personally, I am not a big fan of the name of the method, as well as the resulting acronym. The name "Learning latent causal dynamics" is too general, could be applied to a whole range of methods and does not describe the essence of the proposed method, and the acronym is gathering scattered letters with non-matching capitalization.

---

> ### Author Response · Authors · 2022-08-02
> **Thanks for the review – motivation and discussion of assumptions (Part I)**
>
> Dear Reviewer 7swA,
>
> We greatly appreciate your thorough and constructive comments, many of which have helped improve our paper. Both the paper and appendix have been updated extensively to include as much detail as possible. Please see our point-to-point response to your concerns below.
>
> > Q1: In my eyes, the biggest weakness of this paper is that it does not provide any convincing motivation for the proposed approach. In the current form, the paper argues that existing approaches have a certain set of assumptions and that the proposed approach uses different assumptions. However, this is not really an argument in itself.
>
> A1: Thanks for your comment and suggestion. We have re-written the introduction. We first include a motivating example, i.e., image pixels in videos, as a way to inspire our model setting and argue for those assumptions. In light of your comments, we’ve incorporated one-to-one mappings of the assumptions to their consequential limitations in the updated introduction (L37-L44). We’ve also included how our approach enables recovery of nonparametric latent causal processes under stationary and nonstatioanry environments.  We further respond to the detailed comments below.
>
> > Q1A: What I would like to know is: why do these assumptions matter; how are existing approaches limited by the assumptions that they use; what new applications do you enable with your new approach? To embed these answers into the paper, I would propose a rewrite of the introduction.
>
> A1A: Thanks for the question. We discuss them more explicitly in the updated introduction, which is quoted below.
>
> **Why do these assumptions matter**
> > Estimating latent causal structure from observations, which we assume are unknown (but invertible) nonlinear mixtures of the latent processes, is very challenging. It has been found in [4,5] that without exploiting an appropriate class of assumptions in estimation, the latent variables are not identifiable or disentangled in the most general case. As a result, one cannot make causal claims on the recovered relations in the latent space.
>
> **How are existing approaches limited by the assumptions that they use; what new applications do you enable with your new approach?**
> > To establish identifiability, the existing approaches enforce different sets of functional and distributional form assumptions as constraints in estimation; for example, (1) PCL, GCL, HM-NLICA, and SlowVAE assume mutually-independent sources in the data generating process. However, this assumption may severely distort the identifiability if the latent variables have time-delayed causal relations in between (i.e., causally-related process); (2) SlowVAE and SNICA assume linear relations, which may distort the identifiability results if the underlying transitions are nonlinear, and (3) SlowVAE assumes that the process noise is drawn from Laplacian distribution; i-VAE assumes that the conditional transition distribution is part of the exponential family. However, in real-world scenarios, one cannot choose a proper set of functional and distributional form assumptions without knowing in advance the parametric forms of the latent temporal processes.
>
> > With the proposed condition, our approach allows recovery of latent temporal **causally-related processes** in **stationary environments** without knowing their parametric forms in advance.
>
> > For instance, LEAP established the identifiability of latent temporal processes, but in limited nonstationary cases, under the condition that the distribution of the noise terms of the latent processes varies across segments.
>
> > Accordingly, our approach enables the recovery of latent temporal causal processes in a general nonstationary environment with **time-varying relations** such as changes in the influencing strength or switching some edges off over time or domains.
>
> In case you find anything unclear in the paper, please kindly let us know.
>
> > Q1B: Additionally, it would be nice to include a discussion on the assumptions in section 3.1, i.e. why they are needed and how restrictive they are in practice.
>
> A1B: In light of your comments, we’ve added a new section in the Appendix (S1.4  – Discussion of the Assumptions) to discuss why they are needed, how mild/restrictive these assumptions are, and when they hold in reality.

---

> > ### Author Response · Authors · 2022-08-02
> > **Thanks for the review – problem formulation and discussion of limitations (Part II)**
> >
> > > Q1C: Besides this, the paper is also missing a discussion on the proposed separation of distribution changes into three types. Why did you choose these three types; are there any alternative separations that you considered and why is the proposed solution the best; are there any distribution changes that these types cannot (jointly) represent?
> >
> > A1C: Thank you for pointing it out. We have updated the model formulation in the revised draft to make the logical connections of the settings clearer. Now we start with a fundamental case where we just have stationary time series, which is a regular time series model. We establish the identifiability results in this fundamental case.  Note that the stationary process is not a required component in the latent space, but can actually be thrown away (for even better results). We further consider two cases, i.e., changing causal dynamics, observation changes,  where the stationarity assumptions are violated, and show that the violation of stationarity in both ways can even further improve the identifiability results. This way, we can see how we can deal with different problem settings with our approach. Please refer to L123 - L162 in the updated draft.
> >
> > > Q1D: Finally, a discussion on the limitations of the proposed approach is missing. Overall, this makes it very difficult to judge the significance of this work.
> >
> > A1D: Thanks for pointing it out! In the original submission, we used one sentence to summarize the major limitation (now in L379 in the updated draft).
> >
> > > Extending our theories and framework to address the issue of instantaneous dependency or instantaneous causal relations will be one line of our future work.
> >
> > In light of your comments, we have made the discussion of the limitations more explicit in L370-L378 in the updated draft, which is quoted below.
> >
> > > The basic limitation of this work is that the underlying latent processes are assumed to have no instantaneous causal relations but only time-delayed influences. If the time resolution of the observed time series is the same as the causal frequency (or even higher), this assumption naturally holds. However, if the resolution of the time series is much lower, then it is usually violated and one has to find a way to deal with instantaneous causal relations.
> >
> > Reviewer **nJM9**: Q: “More generally, how do you make evaluate if your data and model satisfies your assumption in the end ?”
> >
> >
> > Furthermore, we would like to mention that our assumed conditions are generally testable. One can run experiments with multiple seeds and compute the similarity (MCCs) between the learned representations; if the representation is not unique, then our assumptions must be violated. We’ve added this discussion immediately after discussing the limitations.
> >
> > > On the other hand, it is worth mentioning that the assumptions are generally testable; if the assumptions are actually violated, one can see that the results produced by our method may not be reliable.

---

> > > ### Author Response · Authors · 2022-08-02
> > > **Thanks for the review – connections to causal discovery and some clarifications (Part III)**
> > >
> > > > Q2A: The connection between this method and causal discovery seems somewhat hand-wavy. The abstract claims that the proposed framework recovers latent variables and identifies their relations. While the experiments and theoretical results support the former claim, the latter seems to be missing from the paper, effectively removing the connection to causal discovery.
> > >
> > > A2A: If one recovers the true latent processes, then identifying their time-delayed causal relations is a classical problem (for instance, in the linear case, Granger causal analysis aims to find such relations). In the first two experiments (simulated dataset and cartpole), we did not report the recovered relations because the ground-truth causal graph is fully-connected; the high MCCs have already indicated the identifiability of the latent causal processes and their causal relations. For the third experiment (CMU-Mocap), we reported the discovered causal graph in Figure 5(d) in the original submission; now in Figure 4(d).
> > >
> > > In light of your comment, we’ve refined Section 2.2 to explain why the latent time-delayed causal relations are immediately discovered once the latent processes can be identified at least up to permutation and component-wise invertible transformations, which are quoted below.
> > >
> > > > Furthermore, if the estimated latent processes can be identified at least up to permutation and component-wise invertible nonlinearities, the latent causal relations are also immediately identifiable because conditional independence relations fully characterize time-delayed causal relations in a time-delayed causally sufficient system,  in which there are no latent causal confounders in the (latent) causal processes. Note that invertible component-wise transformations on latent causal processes do not change their conditional independence relations.
> > >
> > > We’ve also added a new section (Section 4.2 – Causal Visualization)  in the updated draft to explain how latent causal relations can be visualized after the latent processes are recovered.
> > >
> > > > "For visualization purposes, when the underlying latent processes have sparse causal relations, we fit LassoNet on the latent processes recovered by LCD-NM (the acronym has been changed in light of your comment) to interpret the causal relations. Specifically, we fit LassoNet to predict $\hat{\mathbf{z}}\_t$ using the estimated history information $\hat{\mathbf{z}}\_{\text{Hx}} = \\{\hat{\mathbf{z}}\_{t-\tau}\\}_{\tau=1}^{L}$ up to maximum time lag $L$. Note that this postprocessing step is optional – the latent causal relations have already been captured in the learned transition functions in LCD-NM. Also, our identifiability conditions do not rely on the sparsity of causal relations in the latent processes."
> > >
> > > > Q2B: Additionally, "causal dynamics" are mentioned throughout the paper without ever being defined.
> > >
> > > A2B: Thanks for spotting this! We now give an explicit definition of “causal dynamics”, which is quoted below.
> > > > We use "causal dynamics" and "latent causal relations/influences" interchangeably.
> > >
> > > > Q3: According to l.297, all evaluations were conducted on the validation datasets.
> > >
> > > A3: Thanks for the comment – we believe this is a naming issue. We split the datasets into training and validation sets. We did not use the ground-truth latent variables or MCCs in the validation set to tune the hyperparameters of our model. In light of your comments, we’ve changed all the wordings of ``validation set’’ into ‘’test set’’ to avoid possible confusion.
> > >
> > > > Q4A: The paper's clarity could be improved in certain regards. For example, I found it confusing which variables and functions are estimated by the model and which ones are ground-truth factors.
> > >
> > > A4A: Thanks for the question. As stated in Section 3.2, all the variables and functions with a hat symbol are estimated by the model from data, and the ones without a hat symbol are true. Generally, when we describe the data generating process, we use the true variables and functions. We use the estimated variables and functions to describe the estimation procedure. We‘ve made the definition more explicit in the footnote (L84)  in the revised draft.
> > >
> > > > Q4B: Does the model use the ground-truth transition functions, or does it estimate them?
> > >
> > > A4B: We estimate the transition functions and latent causal processes from data without using the ground-truth transition function at all in the estimation. This is one of the main contributions of this study.

---

> > > > ### Author Response · Authors · 2022-08-02
> > > > **Thanks for the review – presentation and some additional information (Part IV)**
> > > >
> > > > > Q4C: Does the model work with unknown distribution shifts as claimed in the abstract and intro, or does it require a domain index (l. 281) as input?
> > > >
> > > > A4C: We do not know precisely how the distribution shifts. However, the current method needs to know the surrogate variable (e.g., domain index) to indicate domain shifts may happen. Without such information, the problem will be more challenging. For instance, one may have to further have a higher-level model for when and how the distribution shifts might happened (for instance, one may use a hidden Markov process for the theta variables in our model, which allows to discovery when and how the distribution shifts happen). If there are real problems following this setting, we will happy to to extend our study to that setting.  In light of your comment, we’ve made the definition clearer in the updated draft (L66-71; L80-81).
> > > >
> > > > > Q5:  Currently, the introduction also functions as a related works section. I would prefer a separation of these two sections. For one, this would allow the introduction to focus on the motivation for the proposed approach, as mentioned above. Additionally, a separate related work section could discuss existing work in more detail and provide an embedding of the proposed work into the existing research landscape - an aspect that is largely missing in the current form.
> > > >
> > > > A5: Thanks for the suggestion! We’ve included a separate “related work” section in the updated draft; the introduction has been rewritten to focus on the motivation.
> > > >
> > > > > Q6: There are grammatical errors and spelling mistakes throughout the paper that a standard spell checker should be able to catch.
> > > >
> > > > A6: Thank you very much for reading our paper carefully. We have carefully edited the updated paper multiple times in order to improve the presentations. We hope you will find the revised paper pleasant to read.
> > > >
> > > > > Q7: Personally, I am not a big fan of the name of the method, as well as the resulting acronym. The name "Learning latent causal dynamics" is too general, could be applied to a whole range of methods and does not describe the essence of the proposed method, and the acronym is gathering scattered letters with non-matching capitalization.
> > > >
> > > > A7: We now make the model name more specific. We call it “learning Latent Causal Dynamics from Nonlinear Mixtures” (LCD-NM). Your further feedback will be extremely appreciated.
> > > >
> > > > With best regards,
> > > >
> > > > Authors of submission 5164

---

> > > > > ### Author Response · Authors · 2022-08-06
> > > > > **Could you kindly check whether our response and updated paper properly addressed your concern?**
> > > > >
> > > > > Dear Reviewer 7swA,
> > > > >
> > > > > Once again, we appreciate your time devoted to reviewing this paper.  We have provided responses to your comments and an updated submission. Could you please check whether they properly addressed your concern? Your feedback would be appreciated. Please kindly let us know in case there are other concerns--we hope we will have the opportunity to respond to them.  Thank you very much!
> > > > >
> > > > > Authors of Paper 5164

---

> > > > > > ### Comment · Reviewer_7swA · 2022-08-08
> > > > > > **Reply**
> > > > > >
> > > > > > Thank you for your extensive reply! I think the revised version of the paper addresses most of my concerns very well, and I will update my rating for this paper accordingly.
> > > > > >
> > > > > > The only part that I am not quite satisfied with is the discussion of the limitations of the proposed method. Currently, the only limitation that is being discussed is that the approach does not work under instantaneous effect. However, like any other model, LCD-NM works with a set of assumptions that need to hold for the method to work - and assumptions always come with limitations. In the current form, the paper includes a discussion on these in the appendix, and I would suggest moving a shortened version of this discussion into the main paper (given the additional space for the camera-ready version). Besides this, I think the paper could be more transparent about its limitations by mentioning the required assumptions in the introduction. Currently, the introduction only discusses the assumptions that other methods require and which can be discarded with the proposed approach. In my eyes, it would be nice to also mention the set of assumptions that LCD-NM needs to work (and how they are less restrictive).

---

> > > > > > > ### Author Response · Authors · 2022-08-08
> > > > > > > **Thanks for updating the score and further feedback**
> > > > > > >
> > > > > > > Dear Reviewer 7eEa,
> > > > > > >
> > > > > > > Thanks for your recognition of our efforts and for kindly updating the score. We are very delighted to hear that most of your significant concerns have been resolved.
> > > > > > >
> > > > > > > In the camera-ready version with an extra page, in light of your further feedback, we will include in the introduction a summary of the required assumptions of our approach (when they apply/are violated in the reality with examples; how they are less restrictive). In the conclusion, we will place a shortened version of the discussion of assumptions and their consequential limitations, together with the limitations pointed out by the other reviewers.
> > > > > > >
> > > > > > > With best regards,
> > > > > > >
> > > > > > > Authors of Paper 5164

---

### Official Review · Reviewer_BcGf · 2022-07-04

**Rating:** 6
**Confidence:** 3
**Soundness:** 3 good
**Presentation:** 2 fair
**Contribution:** 3 good

**Summary:**

This paper proposes a novel method for identifying causal variables from time-series data with time delayed effects. In order to do so, the authors consider data recorded from different distribution shifts/environments. The latent variables are grouped into three main parts: a dynamics part that is fixed across all environments, a dynamics part changing across environments, and finally global changes to the observation space. The paper discusses theoretical results and a practical implementation called LiLY, which outperforms previous methods on a set of synthetic and real-world inspired datasets.

**Questions:**

* Can you provide a clearer comparison of your method to [Huang et al., 2022]?
* What are the limitations of your method from your perspective?

**Limitations:**

As mentioned in the "Weakness" section, no limitations have been discussed, as well as no potential negative social impacts.

**Strengths And Weaknesses:**

## Strengths

The paper proposes an interesting approach to identifying latent causal variables from temporal data. It covers more settings than previous work and supports different transition dynamics and observation noises. The overall approach towards identifiability is clear and closely follows previous works on non-linear ICA. I especially appreciate the thorough discussion in Section 3.1 of various settings which could be identified only with or without the linear independence condition. The theory seems sound, although I have not checked the proofs in the Appendix in detail. The proposed implementation model supposedly supports non-trivial distributions by using Normalizing Flows in the VAE prior. Finally, the experiments cover several settings well in which the proposed model is compared to a set of relevant baselines and shows to outperform previous methods.

## Weaknesses

The clarity of the paper is adversely affected by a confusing notation in several places. Specifically, the following notation was not fully introduced before usage:
* In lines 155-159, the latent space $z$ is indexed by $k$, which had been introduced in Section 2 as a number over domains ($k=1,...,K$) and index of $u$. However, here it seems to be running over different values without explanation.
* Eq 1: the variable "$\tau$" is undefined
* Line 168: the variable "$n$" is undefined
* Between line 224-225 (line numbering issue): the index $r$ as well as Eq. 11 are not in the main text
* Between line 224-225: it is unclear why the different environments are now indexed with $1,...,m$ while previously used $1,...,K$
* Line 225: $\eta_{kt}$ had been defined as $\log p(z_{k,t}|z_{t-1})$, unclear how it is now a function of $u$
* Line 256-260: "$k$" is here again used for domains instead of latent variables
It is strongly recommended to keep the notation consistent throughout the paper, since it is currently difficult to fully understand all statements when the notation is unclear.

A second weakness of the paper is that the comparison to one of the most related works, [Huang et al., 2022] (cited as [33] in the paper), is very sparse. Section 2 beings with 'Following [32,33], ...'. It is unclear what parts are taken from [Huang et al., 2022], and which parts are own definitions/setting descriptions. A more direct discussions of the similarities between this work and [Huang et al., 2022] should be stated. Furthermore, I would have expected a similar discussion in the model part, Section 4, since both follow a similar strategy with an observation setting $\theta^{\text{obs}}$ and dynamics setting $\theta^{\text{dyn}}$.

Another weakness of this work is that it does not discuss the limitations of the work, despite the authors stating so in the paper checklist. Every work on identifiability has its limitations, since certain assumptions need to be taken. However, it is important to address when these assumptions might hold in a realistic setting, and what pre-knowledge was used in creating the models. For instance, Section 4 suggests that in order to obtain identifiability, one needs to exactly know how many variables there are for all three parts: static dynamics, changing dynamics, and observation. This seems quite limiting, since one likely can only estimate those in a real-world setup. How sensitive is the method to this? What happens when too large sets are used? How sensitive is the model to hyperparameters? A fair discussion of other limitations would also be needed. For example, in a real-world system where we do not know the causal variables, how can one ensure that the assumption of Theorem 1 holds (i.e. the $2n$ vector functions being linearly independent)?

Finally, while the experiments cover a decent range of settings, all of them have a rather simple observation space. To strengthen the paper, it is beneficial to include an experiment on a visually complex dataset, for example, Causal3DIdent [von Kügelgen et al., 2021; Zimmermann et al., 2021], more specifically its extension to temporally data [Lippe et al., 2022]. This would highlight the method's capability of identifying causal variables in complex settings.


### Minor points

* Figure 2: the noise variables of the observation, $\hat{\epsilon}\_{i,t}$, are labeled as time-delayed information, although they have the same index and are likely not delayed information.
* The paragraph about the factor representation (line 280-283) comes a bit late, since these have been already used in the previous paragraphs. It would be easier to state this up front.

### Typos

* Line 101: space before comma - "...domains , ..."
* Line 112: "Let u denote the..." (no s)
* Line 184: dot after 1 - "k=1.,2,..."
* Line 185: incomplete word - "So th linear..."
* Line 232: time index $t$ missing - $z_{-1}$
* Line 268: ")" too much
* Line 296: capital "We" within the sentence

### Conclusion

The proposed method is interesting and potentially relevant for researchers in causal representation learning. The current version lacks a bit in clarity and misses a fair discussion of its limitations. Still, I believe that these points can be addressed in a revised version/the rebuttal. Hence, my current score is "Weak Accept".

### References

[Huang et al., 2022] Huang, Biwei, Fan Feng, Chaochao Lu, Sara Magliacane, and Kun Zhang. 2022. AdaRL: What, where, and how to adapt in transfer reinforcement learning. International Conference on Learning Representation (ICLR).

[von Kügelgen et al., 2021] Julius von Kügelgen, Yash Sharma, Luigi Gresele, Wieland Brendel, Bernhard Schölkopf, Michel Besserve, and Francesco Locatello. 2021. Self-Supervised Learning with Data Augmentations Provably Isolates Content from Style. In Thirty-Fifth Conference on Neural Information Processing Systems.

[Zimmermann et al., 2021] Roland S. Zimmermann, Yash Sharma, Steffen Schneider, Matthias Bethge, and Wieland Brendel. 2021. Contrastive Learning Inverts the Data Generating Process. In Proceedings of the 38th International Conference on Machine Learning.

[Lippe et al., 2022] Phillip Lippe, Sara Magliacane, Sindy Löwe, Yuki M. Asano, Taco Cohen, and Efstratios Gavves. 2022. CITRIS: Causal Identifiability from Temporal Intervened Sequences. In Proceedings of the 39th International Conference on Machine Learning.

---

> ### Author Response · Authors · 2022-08-02
> **Thanks for the review – comparisons with AdaRL and discussion of limitations (Part I)**
>
> Dear Reviewer BcGf,
>
> We sincerely thank the reviewer for the time dedicated to reviewing this paper and the helpful comments. Below we give a point-by-point response to the comments.
>
> > Q1: The clarity of the paper is adversely affected by a confusing notation in several places.
>
> A1: We are extremely grateful to you for carefully reading the paper and kindly pointing out the confusing notations. We have edited the paper according to your great observations. Now the we use $r$ to index the domains or regimes and there are $m$ regimes, i.e., $u_r$ with $r=1,2,...,m$. We explicitly define $k$ as the element index of the latent space $\mathbf{z}\_t$ at the beginning of Section 4. The presentations of the other notations have been made clearer.
>
> > Q2: A second weakness of the paper is that the comparison to one of the most related works, [Huang et al., 2022] (cited as [33] in the paper), is very sparse. Section 2 beings with 'Following [32,33], ...'. It is unclear what parts are taken from [Huang et al., 2022], and which parts are own definitions/setting descriptions. A more direct discussions of the similarities between this work and [Huang et al., 2022] should be stated. Furthermore, I would have expected a similar discussion in the model part, Section 4, since both follow a similar strategy with an observation setting θobs and dynamics setting θdyn.
>
> A2: Thank you for your suggestion! We have updated the model formulation in the revised draft. Now we start with a fundamental setting where we have stationary time series, which is a regular time series model. We establish the identifiability results in this fundamental case under mild assumptions. The fundamental setting formulation is orthogonal to the minimal change representation proposed in [Huang et al., 2022].
>
> The other two settings to violate the stationarity assumption consider changes. We introduce the minimal change factors proposed in [Huang et al., 2022]; the nonstationary time series model formulation is in line with [Huang et al., 2022]. Please refer to L122 - L162.
>
> In terms of the implementation, in our work, we enforce the independent noise or conditional independence condition explicitly in Eq. 9 (derived in Appendix S3.1) for the identifiability of the latent processes. But disentanglement is not the main goal of AdaRL, so it used Mixture Density Network (MDN) to approximate the transition prior. Our framework is simpler than AdaRL in the inference module (we use only $\mathbf{x}\_t$ to infer $\mathbf{z}\_t$) and the loss function (we don’t have the prediction branch or the sparsity loss), thanks to the nonlinear ICA formulation in Eq. 1. Our identifiability conditions do not rely on the sparsity constraints in the underlying data generating process. Due to the space limit during rebuttal, we added the discussion of the model part in Appendix S3.2; we will discuss it in Section 5.1 (previously Sec. 4) in the final version.
>
> > Q3. Another weakness of this work is that it does not discuss the limitations of the work, despite the authors stating so in the paper checklist.  Every work on identifiability has its limitations, since certain assumptions need to be taken. However, it is important to address when these assumptions might hold in a realistic setting, and what pre-knowledge was used in creating the models.
>
> A3. Thanks for pointing it out! In the original submission, we used one sentence to summarize the major limitation (now in L379 in the updated draft).
>
> > Extending our theories and framework to address the issue of instantaneous dependency or instantaneous causal relations will be one line of our future work.
>
> In light of your comments, we have made the discussion of the limitations more explicit in L370-L378 in the updated draft, which is quoted below.
>
> > The basic limitation of this work is that the underlying latent processes are assumed to have no instantaneous causal relations but only time-delayed influences. If the time resolution of the observed time series is the same as the causal frequency (or even higher), this assumption naturally holds. However, if the resolution of the time series is much lower, then it is usually violated and one has to find a way to deal with instantaneous causal relations.
>
> Reviewer **nJM9**: Q: “More generally, how do you make evaluate if your data and model satisfies your assumption in the end ?”
>
>
> Furthermore, we would like to mention that our assumed conditions are generally testable. One can run experiments with multiple seeds and compute the similarity (MCCs) between the learned representations; if the representation is not unique, then our assumptions must be violated. We’ve added this discussion immediately after discussing the limitations.
>
> > On the other hand, it is worth mentioning that the assumptions are generally testable; if the assumptions are actually violated, one can see that the results produced by our method may not be reliable.

---

> > ### Author Response · Authors · 2022-08-02
> > **Thanks for the review – discussion of limitations and some additional information (Part II)**
> >
> > > Q4. For instance, Section 4 suggests that in order to obtain identifiability, one needs to exactly know how many variables there are for all three parts: static dynamics, changing dynamics, and observation. This seems quite limiting, since one likely can only estimate those in a real-world setup.
> >
> > A4: In previous Section 4 (now Section 5), we used the initial values of the number of latent variables. But we have criteria to select nontrivial, meaningful causal latent variables. Actually, we always initialized the latent size of each group with a relatively large number. If the initialized latent size is too small, our method may fail. In the end, the extra estimate latent processes will become white noise, in the sense that they do not have temporal dependence and are independent from the remaining meaningful latent processes. Hence, we can choose the number of meaningful latents and select the right ones.
> >
> > > Q5: How sensitive is the method to this?  What happens when too large sets are used? How sensitive is the model to hyperparameters?
> >
> > A5: For instance, in the modified-cartpile, we used 8 latent variables for estimation: 2 of them have temporal structures but the remaining 6 do not have temporal dependencies and are independent from the rest. So the 2 meaningful latent processes were selected. Refer to source code.
> >
> > > Q6: A fair discussion of other limitations would also be needed. For example, in a real-world system where we do not know the causal variables, how can one ensure that the assumption of Theorem 1 holds (i.e. the 2n vector functions being linearly independent)?
> >
> > A6: This is a great question! We cannot ensure this assumption before we get the results. However, if this assumption is violated, the learned latent processes may not be unique. With different initializations, we can find two solutions with the same or similar value of the training objective (although hard). If the assumptions are violated, we will not have unique representations – the two solutions are not permutated componentwise transformations of each other.
> >
> > > Q7: Finally, while the experiments cover a decent range of settings, all of them have a rather simple observation space. To strengthen the paper, it is beneficial to include an experiment on a visually complex dataset, for example, Causal3DIdent [von Kügelgen et al., 2021; Zimmermann et al., 2021], more specifically its extension to temporally data [Lippe et al., 2022]. This would highlight the method's capability of identifying causal variables in complex settings.
> >
> > A7: Thanks for the great suggestion! We are currently running experiments on tempCausal3DIdent. Specifically, we are doing an extension to this temporal dataset without intervention target information. We will let you know once we have the experiment results.
> >
> > > Q8: Figure 2: the noise variables of the observation, ϵ^i,t, are labeled as time-delayed information, although they have the same index and are likely not delayed information.
> >
> > A8: Thank you for the feedback on the presentation. We've labeled $\epsilon^{i,t}$ as observation error in the updated Fig. 2.
> >
> > > Q9: The paragraph about the factor representation (line 280-283) comes a bit late, since these have been already used in the previous paragraphs. It would be easier to state this up front.
> >
> > A9: Thanks for the suggestion! The paragraph “Change Factor Representation” now comes earlier than “Modular Prior Network”.
> >
> > > Q10: Typos. Line 101: space before comma - "...domains , ..." Line 112: "Let u denote the..." (no s) Line 184: dot after 1 - "k=1.,2,..." Line 185: incomplete word - "So th linear..." Line 232: time index t missing - z−1 Line 268: ")" too much Line 296: capital "We" within the sentence.
> >
> > A10: We are extremely grateful to you for carefully reading the paper! We’ve fixed them accordingly.
> >
> > With best regards,
> >
> > Authors of submission 5164

---

> > > ### Comment · Reviewer_BcGf · 2022-08-06
> > > **Rebuttal response**
> > >
> > > I would like to thank the authors for answering my questions in their rebuttal and addressing the presentation issues. Overall, the new version seems more accessible and have a consistent notation. Based on the rebuttal and reading the other reviews, I remain with an overall positive impression and hence support my original recommendation of accepting the paper. Nonetheless, a short follow-up on some aspects:
> > >
> > > Q2: Thank you for clarifying the differences with AdaRL. For a potential camera-ready version with 10 pages, this discussion has to be included in the main paper to fairly compare your method to it.
> > >
> > > Q3/Q6: While I appreciate seeing more effort in the limitations part, it only focuses on instantaneous effects. I highly encourage the authors to discuss further limitations in the final versions, as also pointed out by other reviewers, such as assumptions not being testable, what happens if certain assumptions fail, etc. The authors should not see the limitation section as criticizing their own work, but rather fairly put it into perspective for potential use cases.
> > >
> > > Q7: It is great to hear that you are running additional experiments on this dataset, I would be curious about the results. Note that I will not consider the performance of the method on that dataset as a criterion for recommended acceptance/rejection here, but rather would like to see that the authors test out the boundaries of their method. It is even fair if the method fails on the dataset, it would then be clearer on what sets of datasets the method may be applicable.

---

> > > > ### Author Response · Authors · 2022-08-08
> > > > **Thanks for recommendation of accepting the paper and further comments**
> > > >
> > > > Dear Reviewer BcGf,
> > > >
> > > > Thank you for your encouraging feedback and recommendation of accepting the paper! In the camera-ready version (with an extra page), we will absolutely embed the model comparison with AdaRL in "Section 5 – Our Approach" together with the description of our model. We will also use the extra page to include the limitations pointed out by the other reviewers. Finally, we would like to let you know that, in the final version, we will include the results of our model on the original tempCausal3DIdent, as well as a modified version that is more suitable for our approach (without intervention target information; only domain index). We are actively contacting the authors for license permission and access.
> > > >
> > > >
> > > > Thank you,
> > > > Authors of submission 5164

---

### Official Review · Reviewer_nJM9 · 2022-07-11

**Rating:** 6
**Confidence:** 3
**Soundness:** 3 good
**Presentation:** 2 fair
**Contribution:** 2 fair

**Summary:**

This paper proposes a formulation to recover latent temporal processes from observed time series data. Assumptions for identifability of the latent processes relax the ones posited in previous works and now rely on linear independence of specific vectors of partial derivatives of the latent update function. Authors also consider the case with observations from different domains and with time varying causal effects between hidden processes. The authors then propose an architecture enforcing the specific linear independence assumption and show that they can identify the latent temporal processes for an artificially generated and cart pole dataset.

**Questions:**

- What are the exact conditions on g(.) ? Should it be an invertible function ?
- Enforcing conditions of Eq 4. With prior model and KL loss. Use sampling to infer KL divergence. How do you make sure that this results in satisfying conditions of Eq.4 ? More generally, how do you make evaluate if your data and model satisfies your assumption in the end ? Was there any issue in the training and in getting the KL loss low ?
- How does the sampling for the KL loss impacts computational considerations. How does this scale for larger number of latents and number of lags?
- How do you divided the number of latents for each group (fixed,c,o) ? What happens when the number of latents is misspecified ? It seems you use n=8 in the cart pole but the true number of latents is 2. How do you select the right ones ?

**Limitations:**

I think the assumptions of the model are clear in general, except for the read-out function g(.) (see above).


**Strengths And Weaknesses:**

Strengths :

- The problem addressed is important and builds upon solid previous work.
- The work relaxes distributional assumptions from previous works and provides identifiability proof.
- The reported performance are clearly above other compared methods.

Weaknesses :

- There are some clarifications needed that I listed as questions below.
- Why didn’t you evaluate the MCC of other baselines on the cart pole dataset ? This seems a more informative experiment for comparing the methods than the synthetic data that exactly matches your data generating assumption.
- It would be great to have a formulation of the identifiability criteria for arbitrary lags. And understand how it impacts training.
- Overall I’m not really convinced by the causal taxonomy of the model. Some assumptions are posited but it does not guarantee any causal discovery mechanism. The model formulation doesn't also allow to draw any causal conclusion between observables. In particular, it feels that the action in the cart pole data is modelled as a different set of parameters which thus don’t really model the treatment effect in the observed process.

Minor
- C is not shown on Figure 2
- Some typos :
    - Line 49 : question arises *that* (does not sound really correct to me)
    - Line 185 : So *th* linear independence (« the » ?)

---

> ### Author Response · Authors · 2022-08-02
> **Thanks for the review - clarifications and some additional information (Part I)**
>
> Dear Reviewer nJM9,
>
> We appreciate that you read our paper very carefully and your informative feedback, which has helped improve our paper. Below please see our response to your concerns. In case you find anything unclear in the paper, please kindly let us know.
>
> > Q1: Why didn’t you evaluate the MCC of other baselines on the cart pole dataset? This seems a more informative experiment for comparing the methods than the synthetic data that exactly matches your data generating assumption.
>
> A1: Thanks for raising the concern. We actually showed the MCC of the baselines which produced the highest MCC values on cartpole. The MCCs of the other baselines are below 0.6 in general. In light of your comment, we’ve included the results for all baselines on the cart pole dataset and updated Figure 3C.
>
> > Q2: It would be great to have a formulation of the identifiability criteria for arbitrary lags. And understand how it impacts training.
>
> A2: Thank you for the suggestion! For the sake of simplicity, we consider one lag for the latent processes in Sec. 4 (and only in Sec. 4). Our identifiability proof can actually be directly applied for arbitrary lags. For instance, in the stationary case in Eq. 4, one can simply re-define $\eta\_{kt} \triangleq \log p(z\_{k, t} | \mathbf{z}\_{Hx})$, where  $\mathbf{z}\_{Hx}$ denotes the lagged latent variables up to maximum time lag $L$. We plug it into Eq. 4, and take derivatives with regard to $z\_{1, t-\tau}, …, z_{n, t-\tau}$, which can be latent temporal variables at any lag index $\tau$, instead of $z\_{1, t-1}, …, z\_{n, t-1}$. If there exists one $\tau$ (out of the $L$ lags) that satisfies the condition, then the stationary latent processes is identifiable. Similarly, for Eq. 8, one can simply re-define $\eta\_{kt}(u\_r)  \triangleq \log p(z\_{k, t} | \mathbf{z}_{Hx}, u_r)$ and plug it into Eq. 8. No extra changes are needed.
> In light of your comments, we’ve included one section in the updated Appendix S.1.5 – Extension to Multiple Time Lags. In terms of its impacts on training, we actually have already used multiple time lags, i.e., $\mathbf{z}\_{Hx}$ in our Section 5 – Our Approach and in the implementaion. So arbitrary time lags won’t have further impacts on the training.

---

> > ### Author Response · Authors · 2022-08-02
> > **Thanks for the review - clarifications on causal taxonomy and some additional information (Part II)**
> >
> > > Q3: Overall I’m not really convinced by the causal taxonomy of the model. Some assumptions are posited but it does not guarantee any causal discovery mechanism. The model formulation doesn't also allow to draw any causal conclusion between observables.
> >
> > A3: Thanks for the comment. First of all, in this work, we consider the time-delayed causal relations in time series without instantaneous relations and use this feature of causal influences to recover the underlying hidden causal processes satisfied with time-delayed (causal) influences. We understand that there is a large body of work that discovers instantaneous causal relations from independent identically distributed (i.i.d.) data or low-resolution data. For the connection between time-delayed causal relations and structural equation models, one may refer to [Fisher, 1970][Dash and Marek, 2001][Gong et al., 2017].
> >
> > Furthermore, although a number of causal discovery methods focused on recovering causal relations among the observed variables (e.g., PC [Spirtes et al., 2000], FCI [Spirtes et al., 2000], or Linear Non-Gaussian Acyclic Model (LiNGAM [Shimizu, et al.,2006])), in many real-world scenarios, such as questionnaire-based psychometric studies, the measured variables do not have direct causal edges but are generated by underlying hidden variables or confounders that are causally related, as discussed by [Silva et al., 2006], and GIN [Xie et al., 2020]. In this work, we consider the situation in which one has to learn the causally-related temporal latent processes from the observed temporal variables. This setting can explain a number of real problems but was clearly under-explored.
> >
> > (If the time-delayed causal relations happen to be among the observables, the learned mixing function $g$ in Eq. 1 will be identity mappings, which is a special case of the nonlinear invertible mixing function, and the causal relations among observables can be captured by transition functions $f_{s}$ or $f_{c}$.)
> >
> > If more details are needed for the causal taxonomy used here, please kindly let us know.
> >
> > ### References
> >
> > [Fisher, 1970] Fisher, Franklin M. "A correspondence principle for simultaneous equation models." Econometrica: Journal of the Econometric Society (1970): 73-92.
> >
> > [Dash and Marek, 2001] Dash, Denver, and Marek Druzdzel. "Caveats for causal reasoning with equilibrium models." European Conference on Symbolic and Quantitative Approaches to Reasoning and Uncertainty. Springer, Berlin, Heidelberg, 2001.
> >
> > [Gong et al., 2017] Gong, Mingming, et al. "Causal discovery from temporally aggregated time series." Uncertainty in artificial intelligence: proceedings of the... conference. Conference on Uncertainty in Artificial Intelligence. Vol. 2017. NIH Public Access, 2017.
> >
> > [Spirtes, et al., 2000] Spirtes, Peter, et al. Causation, prediction, and search. MIT press, 2000.
> >
> > [Shimizu, et al.,2006] Shimizu, Shohei, et al. "A linear non-Gaussian acyclic model for causal discovery." Journal of Machine Learning Research 7.10 (2006).
> >
> > [Silva et al., 2006] Silva, Ricardo, et al. "Learning the Structure of Linear Latent Variable Models." Journal of Machine Learning Research 7.2 (2006).
> >
> > [Xie et al., 2020] Xie, Feng, et al. "Generalized independent noise condition for estimating latent variable causal graphs." Advances in Neural Information Processing Systems 33 (2020): 14891-14902.
> >
> > > Q4: In particular, it feels that the action in the cart pole data is modelled as a different set of parameters which thus don’t really model the treatment effect in the observed process.
> >
> > A4: Indeed the action is modeled as a set of parameters. Our formulation treats the action like a domain index or one change factor $\theta$ variable. Please note that we do not consider cartpole as a control or Reinforcement Learning (RL) problem in our experiments. Instead, we are provided with a constant action value (left or right) and the observed video in each setting. Our goal was to recover the underlying meaningful latent processes (i.e., pole angle, cart position). One may further consider the action as a particular process and develop a specific method to learn meaningful processes in the RL context, which is not our focus in the paper.
> >
> > > Q5: C is not shown on Figure 2. Some typos : Line 49 : question arises that (does not sound really correct to me). Line 185 : So th linear independence (« the » ?)
> >
> > A5: Thank you very much for reading our paper carefully and pointing out the typos! C is the change factor representation module in Figure 2 and we’ve added the text annotation in the updated presentation. We've fixed this typo in the revised draft.

---

> > > ### Author Response · Authors · 2022-08-02
> > > **Thanks for the review - exact conditions on g(.) and training details (Part III)**
> > >
> > > > Q6: What are the exact conditions on g(.) ? Should it be an invertible function ?
> > >
> > > A6: Correct! The exact conditions of g(.) are that g is an invertible function (previously defined in L114; now in L124).
> > >
> > > > Previous L114: In each domain, we observe time series {\mathbf{x}\_t}\_1^T generated by an arbitrary invertible nonlinear mixture of the underlying temporal processes $\mathbf{z}\_t$.
> > >
> > > > Previous L122: …the mixing function $g$ in Eq. 1…
> > >
> > > In light of your comments, we’ve made the connections smoother in the updated draft.
> > >
> > > > Now L124: the observations $\mathbf{x}\_t$ comes from \xrev{a nonlinear (but invertible) mixing function} $\mathbf{g}$ that maps the time-delayed causally-related latent variables $\mathbf{z}\_t$ to $\mathbf{x}\_t$
> > >
> > > > Q7: Enforcing conditions of Eq 4. With prior model and KL loss. Use sampling to infer KL divergence. How do you make sure that this results in satisfying conditions of Eq.4?  More generally, how do you make evaluate if your data and model satisfies your assumption in the end?
> > >
> > > A7: Thanks for raising this practical issue! First, Eq. 4 are  **assumptions over data properties** and do not need to be enforced as constraints in the KL divergence. Furthermore, we would like to mention that the assumptions are generally testable. For instance, if Eq. 4 is violated, then the solution to the problem will not be unique. The learned latent causal processes under different random seeds will not be componentwise transformations of each other. In the updated draft, we’ve added the discussion in L376 immediately after the limitations .
> > >
> > > > Q8: Was there any issue in the training and in getting the KL loss low?
> > >
> > > A8: Thank you for the question. We didn’t encounter such issues. If you would like to see any details on any specific points, please kindly let us know.
> > >
> > > > Q9: How does the sampling for the KL loss impacts computational considerations. How does this scale for larger number of latents and number of lags?
> > >
> > > A9: The KL loss computation scales linearly in time with the latent size and the number of lags. In our experiments, one GPU (11GB memory) is sufficient to perform all the experiments.
> > >
> > > > Q10: How do you divided the number of latents for each group (fixed,c,o)? What happens when the number of latents is misspecified? It seems you use n=8 in the cart pole but the true number of latents is 2. How do you select the right ones?
> > >
> > > A10: Actually, we always initialized the latent size of each group with a relatively large number. If the initialized latent size is too small, our method may fail. The extra estimated latent processes will become white noise in the end, in the sense that they do not have temporal dependence and are independent from the remaining meaningful estimated latent processes. Hence, we can select the right ones.
> > >
> > > **Cartpole**
> > >
> > > You are totally right. We used 8 latent variables for estimation: in the end, 2 of them have temporal structures but the remaining 6 do not have temporal dependencies and are independent from the rest.
> > >
> > > With best regards,
> > >
> > > Authors of submission 5164

---

> > ### Comment · Reviewer_nJM9 · 2022-08-08
> > **Thank you for your response**
> >
> > Dear Authors,
> >
> > Thank you for answering my comments. I have no further questions at this point.
> >
> > Best Regards,

---

> > > ### Author Response · Authors · 2022-08-08
> > > **Would you like to consider updating your recommendation?**
> > >
> > > Dear Reviewer nJM9,
> > >
> > > Thanks for your quick response! Would you like to consider updating your recommendation, if your concerns are properly addressed?
> > >
> > > With best regards,
> > >
> > > Authors of submission 5164

---

> ### Comment · Area_Chair_L8jn · 2022-08-08
> **Please respond to author feedback**
>
> Thank you for reviewing this paper. Could you respond to the author feedback, or at least acknowledge that you've read the reply? Does the author reply address your concerns?
>
> Best, AC

---

### Official Review · Reviewer_U8Ti · 2022-07-11

**Rating:** 5
**Confidence:** 3
**Soundness:** 2 fair
**Presentation:** 3 good
**Contribution:** 2 fair

**Summary:**

This paper considered establishing identifiability theories for nonparametric latent causal processes from their observed nonlinear mixtures. The authors proposed to recover time-delayed latent causal variables and identify their relations from measured temporal data, namely LiLY. The theoretical results were derived for both stationary environments and unknown distribution shifts, with promising results in real-world applications.

**Questions:**

Please check the questions listed in the Weaknesses above. Any justifications are welcomed.

**Limitations:**

I didn't find any limitations discussed. Please correct me if I am missing anything.

**Strengths And Weaknesses:**

*Strengths:*

 - This paper handles an interesting problem in machine learning by considering the identifiability of causal dynamics in time series data.
 - The theoretical analyses are sound with promising real data analysis results.
 - This paper is well written and organized.

*Weaknesses:*

- The theoretical performance of the proposed LiLY for learning latent causal dynamics was not derived and discussed. Can the authors provide any theoretical analysis on this?
- The partitioned latent subspace is pre-specified and limited to three components. Can the authors provide more justification for this?

---

> ### Author Response · Authors · 2022-08-02
> **Thanks for the review - justification and some additional information**
>
> Dear Reviewer U8Ti,
>
> We are sincerely grateful to the reviewer for the informative feedback, which has helped improve our paper (please see the updated paper and appendix.) Please see our point-to-point response below.
>
> > Q1: The theoretical performance of the proposed LiLY for learning latent causal dynamics was not derived and discussed. Can the authors provide any theoretical analysis on this?
>
> A1: Thank you for the question. We provided the theoretical identifiability guarantees of the underlying latent processes, which constitute one type of theoretical analysis in the asymptotic case. Deriving the theoretical error bound of nonlinear ICA on finite data is particularly challenging, and may lead to a separate learning theory paper. However, we would be happy to provide empirical results to illustrate the errors. It would be highly appreciated if you could kindly let us know which line of research is more consistent with what you meant by “theoretical analysis”. Your further feedback will be extremely helpful for refining our understanding.
>
> > Q2: The partitioned latent subspace is pre-specified and limited to three components. Can the authors provide more justification for this?
>
> A2: Thanks for pointing it out! We have updated the model formulation in the revised draft (Sec. 3.1) to make the logical connections of the settings clearer. Now we start with a fundamental case where we just have stationary time series, which is a regular time series model. We establish the identifiability results in this fundamental case.  Note that the stationary process is not a required component in the latent space, but can actually be thrown away (for even better results). We further consider two cases, i.e., changing causal dynamics, observation changes,  where the stationarity assumptions are violated, and show that the violation of stationarity in both ways can even further improve the identifiability results. This way, we can see how we can deal with different problem settings with our approach. Please refer to L122 - L162.
>
> > Q3: Please check the questions listed in the Weaknesses above. Any justifications are welcomed.
>
> A3: If the formulation is presented this way in the updated draft (L122 - L162), we do not start with the 3 components; instead, we consider a fundamental situation and violations of the stationarity assumptions and show how to tackle the problem in different situations.
>
> > Q4: I didn't find any limitations discussed. Please correct me if I am missing anything.
>
> A4: Thanks for pointing it out! In the original submission, we used one sentence to summarize the major limitation (now in L379 in the updated draft).
>
> > Extending our theories and framework to address the issue of instantaneous dependency or instantaneous causal relations will be one line of our future work.
>
> In light of your comments, we have made the discussion of the limitations more explicit in L370-L378 in the updated draft, which is quoted below.
>
> > The basic limitation of this work is that the underlying latent processes are assumed to have no instantaneous causal relations but only time-delayed influences. If the time resolution of the observed time series is the same as the causal frequency (or even higher), this assumption naturally holds. However, if the resolution of the time series is much lower, then it is usually violated and one has to find a way to deal with instantaneous causal relations.
>
> "Reviewer **nJM9** Q: More generally, how do you make evaluate if your data and model satisfies your assumption in the end?”
>
>
> Furthermore, we would like to mention that our assumed conditions are generally testable. One can run experiments with multiple seeds and compute the similarity (MCCs) between the learned representations; if the representation is not unique, then our assumptions must be violated. We’ve added this discussion immediately after discussing the limitations.
>
> > On the other hand, it is worth mentioning that the assumptions are generally testable; if the assumptions are actually violated, one can see that the results produced by our method may not be reliable.
>
> With best regards,
>
> Authors of submission 5164

---

> > ### Author Response · Authors · 2022-08-06
> > **Could you please let us know whether our responses and updated submission properly addressed your concern?**
> >
> > Dear Reviewer U8Ti,
> >
> > Thank you very much for your time spent on our submission and your questions. We have tried to address your concerns in the response and updated submission--any feedback from you would be appreciated. If you have further comments, please kindly let us know--we hope for the opportunity to respond to them.
> >
> > Best wishes,
> > Authors of paper 5164

---

> > > ### Comment · Reviewer_U8Ti · 2022-08-08
> > > **Thank you**
> > >
> > > Thanks for the response. My major concerns have been nicely addressed.

---

> > > > ### Author Response · Authors · 2022-08-08
> > > > **We appreciate your timely feedback**
> > > >
> > > > Dear Reviewer U8Ti,
> > > >
> > > > Thank you very much for examining our response and letting us know that your major concerns have been nicely addressed.  In this case, could you please consider updating your recommendation accordingly?
> > > >
> > > > Best wishes,
> > > > Authors of paper 5164

---

> > > > ### Author Response · Authors · 2022-08-09
> > > > **Could you please consider updating your recommendation, given that your major concerns were nicely addressed?**
> > > >
> > > > Dear Reviewer U8Ti,
> > > >
> > > > We understand you are busy and appreciate your time.  Here we are re-sending a previous message to make sure you see it.  Thank you very much for letting us know that your major concerns have been nicely addressed. In this case, could you please consider updating your recommendation to reflect it (the original rating is 5)?
> > > >
> > > > Your feedback will be highly appreciated.
> > > >
> > > > With best regards,
> > > >
> > > > Authors of paper 5164

---

> ### Comment · Area_Chair_L8jn · 2022-08-08
> **Please respond to author feedback**
>
> Thank you for reviewing this paper. Could you respond to the author feedback, or at least acknowledge that you've read the reply? Does the author reply address your concerns?
>
> Best, AC

---

### Author Response · Authors · 2022-08-04
**Do the response and revision address your concern?**

Dear Reviewers U8Ti, nJM9, BcGf, and 7swA,

Thanks for your time dedicated to carefully reviewing this paper. It would be further highly appreciated if you let us know whether our response and the change in the paper properly address your concerns, despite your busy schedule. Thanks a lot!

With best regards, Authors of submission 5164

---

### Meta-Review · Area_Chair_L8jn · 2022-08-20

**Recommendation:** Accept
**Confidence:** Certain

**Metareview:**

Decision: Accept

This paper presents identifiability results of causal relationships in non-stationary time series under the assumption of latent dynamics with non-linear mixtures. The authors also propose an implementation of the assumed causal model as a sequential deep generative model. Experiments compare the proposed method with existing causal ML methods on time series & sequential VAE and obtained better results.

Reviewers commended the paper as well motivated and agreed that the paper's assumptions are more practical than existing causal ML work on time series data. There were questions on the assumptions & technical results from some reviewers, which were largely addressed in author feedback. I'd also encourage the authors to include some of the replies in author feedback to the camera ready, as this will help clarifying the approach further.

As a side note, some reviewers are not very comfortable with the paper title "Causal Disentanglement for Time Series", as they think the title is perhaps "too ambitious and general to reflect this paper's main focus and disappoints audiences who imagine something else". I'd encourage the authors to consider this comment from the reviewers.

**Award:**

No

---

### Decision · Program_Chairs · 2022-09-14

Accept